# Diversity of the Pacific Ocean coral reef microbiome

Coral reefs are among the most diverse ecosystems on Earth. They support high biodiversity of multicellular organisms that strongly rely on associated microorganisms for health and nutrition. However, the extent of the coral reef microbiome diversity and its distribution at the oceanic basin-scale remains to be explored. Here, we systematically sampled 3 coral morphotypes, 2 fish species, and planktonic communities in 99 reefs from 32 islands across the Pacific Ocean, to assess reef microbiome composition and biogeography. We show a very large richness of reef microorganisms compared to other environments, which extrapolated to all fishes and corals of the Pacific, approximates the current estimated total prokaryotic diversity for the entire Earth. Microbial communities vary among and within the 3 animal biomes (coral, fish, plankton), and geographically. For corals, the cross-ocean patterns of diversity are different from those known for other multicellular organisms. Within each coral morphotype, community composition is always determined by geographic distance first, both at the island and across ocean scale, and then by environment. Our unprecedented sampling effort of coral reef microbiomes, as part of the *Tara* Pacific expedition, provides new insight into the global microbial diversity, the factors driving their distribution, and the biocomplexity of reef ecosystems.

Corals build three-dimensional calcium carbonate skeletons that give rise to the reef framework, providing shelter for a large diversity of fish and other organisms. The global biodiversity of tropical coral reefs is estimated to reach 830,000 species worldwide, which represent 32% of all named marine multicellular species[1]. Coral reef biodiversity is a key component of marine ecosystems and is also critically important to humans who benefit from the reefs' ecological, medicinal, and cultural goods and services[2]. Reefs are, however, globally impacted by climate change and by direct human activity leading to habitat destruction and unprecedented loss of reef cover[3]. These perturbations are so damaging that half of the global coral coverage has disappeared since the 1950s, and the associated species diversity has declined by more than 60%[4], prompting urgency to devise strategies that can reverse or halt the current trend[5,6].

Microorganisms are the invisible yet essential component of coral reefs where they drive and maintain productivity and biodiversity, and

exemplify the essential role of symbiosis[7,8]. Bacteria live in close association with corals and contribute to host physiology and fitness by participating in nutrient acquisition, metabolic (re)cycling, and protection against pathogens[7,9–13]. Similarly, the role of the microbiome is also essential in other emblematic reef animals such as fish[14–17].

Despite the fact that coral reefs represent hotspots of biodiversity on Earth, the global diversity of its associated microorganisms (microbiome) is still poorly estimated, as it has often been assessed only locally and studied separately for different reef organisms. Recent attempts at counting diversity across species reported from 31,000 operational taxonomic units at 97% similarity ($OTUs_{97}$) to 129,000 in Australian reefs[18,19], 44,000 amplicon sequence variants (ASVs) in an Indian Ocean reef[14], and similar estimates for the Red Sea[20]. These are small numbers compared to recent global estimates predicting that as many as 2.72 to 5.44 million distinct prokaryotic ASVs are present on

✉ e-mail: pierre.galand@obs-banyuls.fr

Earth[21]. Global diversity assessments are, however, a matter of debate[22] as they rely on extrapolations that result in numbers/ranges that can vary by orders of magnitude[23,24]. Global surveys of the Earth microbial diversity could provide better estimates, but they are rare and also show very large variations[25–27]. The extent of free-living and host-associated microbial diversity in coral reefs at the ocean scale remains unknown.

For the coral host, patterns of community diversity are well established. The most diverse coral communities are found in the Western Pacific Ocean where the reefs in the coral triangle harbour the highest number of coral species[28]. Community diversity then decreases eastward reaching the lowest values along the Eastern Pacific coast of Central America. Moreover, corals of the Pacific Ocean show clear patterns of biogeography defined by distinct faunal provinces, shaped by long-term historical processes[29], and separated by sharp breaks like the Eastern Pacific Barrier[30]. However, it is not currently known whether the coral-associated microbiome follows similar patterns of diversity and biogeography as the host animal.

In this work we assess the diversity of the Pacific Ocean reef microbiome. To do so, we designed a systematic basin-scale sampling strategy that targeted 3 types of organisms, which fulfil crucial ecological roles on coral reefs: corals, fish, and plankton[31,32]. We conducted an unprecedented campaign to methodically sample the free-living and particle/eukaryote associated planktonic microorganisms, one carnivorous (*Zanclus cornutus*) and one herbivorous (*Acanthurus triostegus*) fish species, and three coral morphotypes that belong to distinct clades (the hydrozoan *Millepora platyphylla*, and the two anthozoans *Porites lobata* and *Pocillopora meandrina*). These species are widespread and among the few that occur across most of the Pacific Ocean[32]. Samples were collected across 99 reefs of the Pacific Ocean, and we analysed more than 5000 microbiomes by means of metabarcoding of the 16 S rDNA V4/V5 region. We show a very large richness of reef microorganisms compared to other environments, and demonstrate that for corals, community composition is always determined by geographic distance first, both at the island and across ocean scale, and then by the environment.

## Results

### Microbial community diversity of the plankton, coral, and fish biomes

In total, we captured 542,399 ASVs from 5,392 samples from the three biomes across 32 islands, comprising close to 3 billion sequences (2.87 billion) (Fig. 1), which is about a quarter more than the entire EMP (2.2 billion sequences from 27,751 samples)[25]. Extrapolation of the microbial richness from our dataset using a fitted Preston model[33] showed that our sequencing effort uncovered overall 98% of the diversity of the biomes we sampled (separately, 98% for coral and plankton, and 90% for fishes).

The accumulation curve based on community richness showed that diversity increased sharply with the addition of samples from each new biome (Fig. 1b). Starting with the most diverse biome of plankton, an exponential increase was followed by a plateau for the small size fraction that corresponds to the free-living microorganisms ( < 3 μm). It rapidly increased again when adding larger plankton size fractions that include microorganisms associated to planktonic eukaryotes or particles (Fig. 1b). A sharp increase was again observed when adding the first coral samples from one morphotype. The addition of subsequent coral and fish genera led to a slower increase in diversity (Supplementary Fig. 1).

Microbial community diversity differed between biomes (coral, fish and plankton) and genera within biomes (Shannon index, Wilcoxon, p < 0.01), with the exception of the hydrocoral *Millepora* vs. the fish *Zanclus cornutus*. Plankton communities, including free-living and attached prokaryotes, were the most diverse. Within corals, the hydrozoan *Millepora* had the highest microbiome diversity followed

by the two scleractinians *Porites* and *Pocillopora*. For fish, *Zanclus cornutus* microbial communities had higher diversity than *Acanthurus triostegus* communities (Fig. 1c). In terms of richness, plankton samples on average comprised 7,100 ASVs (sd ± 552), *Millepora* 3,833 ( ± 396), *Porites* 1,566 ( ± 238), *Pocillopora* 824 ( ± 99), *Zanclus* 964 ( ± 49), and *Acanthurus* 490 ( ± 30).

Microbial community composition was significantly different between the three biomes (Fig. 1d) (PERMANOVA, p < 0.01). Community composition also differed among coral species (Fig. 2), between the two fish species and between the gut and skin microbiomes for each fish species (Fig. 2) (PERMANOVA, $p < 0.01$). Interestingly, gut microbial communities were clearly separated, while the mucus communities partially overlapped (Fig. 2). For both the free-living (0.2–3 μm) and the particle or eukaryote associated (3–20 μm) size fractions of plankton, the communities sampled near the islands were different from the communities sampled at the surface above the coral colonies and close to the *Pocillopora* colonies; the latter two were similar to each other (Fig. 2).

We did not detect any ASVs that were present in all samples across plankton, coral, and fish (Fig. 1e). The ASV with the highest prevalence belonged to the family Vibrionaceae (class Gammaproteobacteria, asv0000004) that was present overall in 94% of the samples, and 98% of the plankton samples, 94% of the coral samples, 92% of the fish samples. The most abundant ASVs belonged to the family Endozoicomonadaceae (asv0000001; 114,034,620 sequences), present overall in 54% of the samples, and 56% of the plankton samples, 80% of the coral samples, 20% of the fish samples.

At the class level, the plankton communities were characterised by a higher proportion of Cyanobacteria and the presence of Thermoplasmata (Archaea) and Acidimicrobiia (Supplementary Fig. 2). For corals, the hydrozoan *Millepora* had more Spirochaetia, while the scleractinian *Porites* had more Chlorobia, and *Pocillopora* more Gammaproteobacteria. Both fish species were characterised by the presence of Clostridia and Bacilli, and *Zanclus* had a higher proportion of Desulfovibrionia (Supplementary Fig. 2).

### Microbial communities of *Millepora*, *Porites*, and *Pocillopora*

For each coral morphotype, 3 sites were sampled at each island and 10 different colonies were collected per site on average, resulting in a dataset of 619 *Millepora*, 945 *Porites*, and 977 *Pocillopora* samples.

Microbial community diversity patterns (Shannon index) across the Pacific Ocean were different among the 3 coral morphotypes (Fig. 3). For *Millepora*, the highest average value was observed at Babeldaob in Palau (I26), then Solomon Islands (I22) and Normanby Island in Papua New Guinea (I23) in the coral triangle, followed by geographically close sites in the South Pacific (Niue (I09), Upolu (I10), Wallis and Futuna (I11)). For *Porites*, the highest microbial diversity was observed at Kiribati (I13) and Crescent Island in Hong Kong (I27), outside the coral triangle, followed by the lowest diversity sites in the southernmost islands (Fig. 3). For *Pocillopora*, the site with highest diversity were dispersed from the South Pacific (Upolu (I10) and Wallis and Futuna (I11)), the coral triangle (Salomon Island (I22)) and the Eastern Pacific (Malpelo (I03)). For all coral morphotypes, microbial community diversity varied between sites for some islands, and *Millepora* and *Porites* generally showed a higher between-site variation in diversity than *Pocillopora* (Fig. 3, Supplementary Fig. 3).

Comparisons of similarity between microbial communities showed that for all 3 coral morphotypes, communities sampled within the same site were more similar to each other than communities sampled at different sites within the same island that, in turn, were more similar than communities from different islands (unpaired Wilcoxon test $p < 0.01$). *Pocillopora* had the highest level of dissimilarity between samples (Fig. 4), which was reflected by an overall higher beta-dispersion in community composition (beta-diversity variance

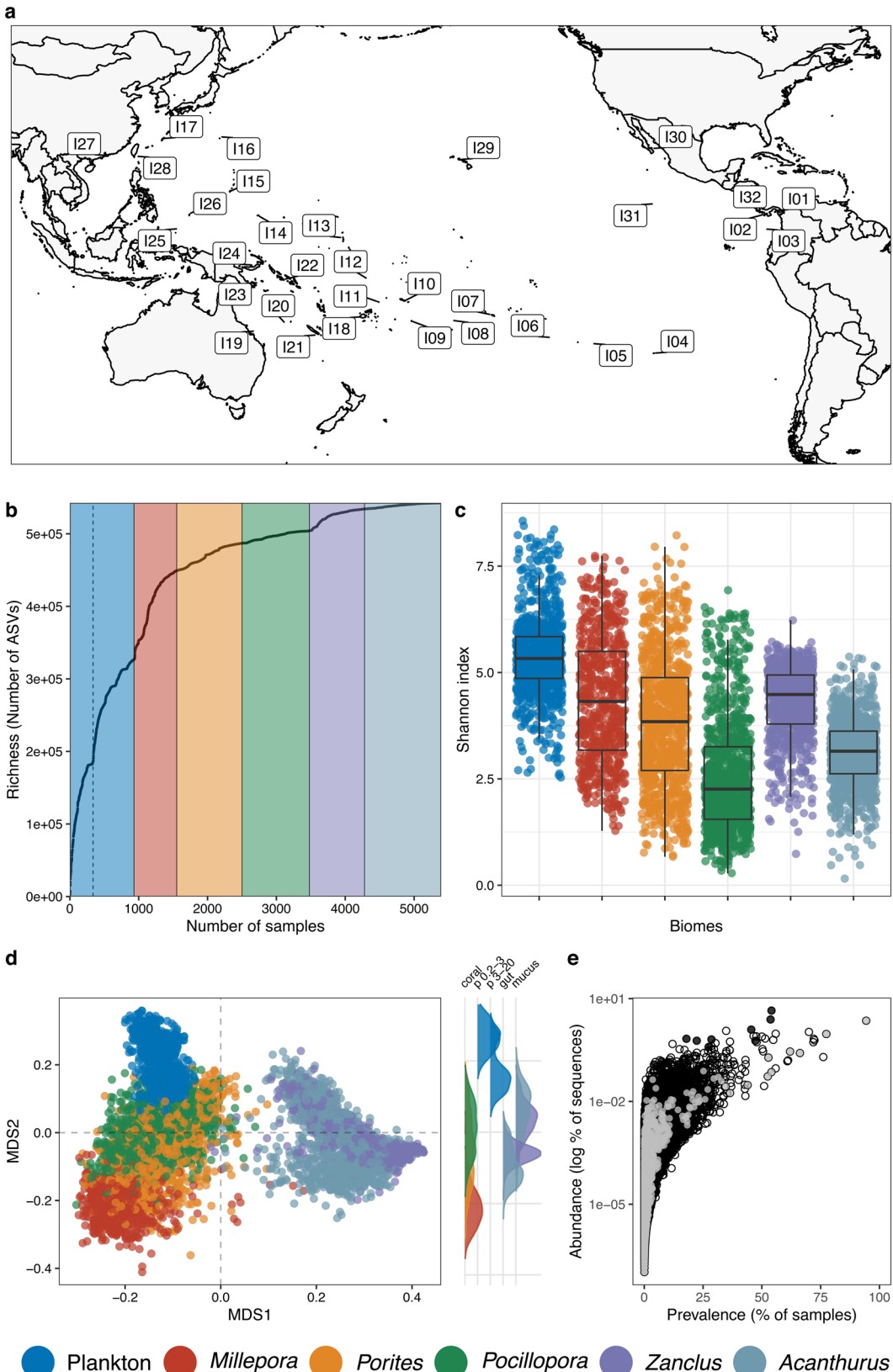

within a group); *Porites* had the lowest beta-dispersion values (Supplementary Fig. 4), but distributions were largely overlapping.

We used hierarchical clustering to group microbial communities according to their composition and to identify similarities in microbial community composition between islands (Fig. 5, Supplementary Fig. 5). The results show that the pattern of microbial community composition across the Pacific Ocean was different between the 3 coral morphotypes (Fig. 5, Fisher test, p = 0.22). Microbial communities of *Millepora* showed patterns of biogeography in which, generally, communities that were closer to each other were more similar (Fig. 5a, Supplementary Fig. 6, Supplementary Fig. 7). For *Porites*, communities from the Eastern Tropical Pacific were similar to each other, but

**Fig. 1 | Diversity and community composition of the plankton, coral, and fish microbiomes across 32 islands of the Pacific Ocean. a** Map of the islands sampled. **b** Accumulation curves of microbial community richness. The dashed line represents the shift between the small planktonic size fraction ( < 3 μm) and the larger size fractions ( > 3 μm). **c** Shannon diversity index across all samples ($n = 3,298$). The box plot horizontal bars show the median value, the box indicates the first and third QRs, and the whiskers indicate 1.5*IQR. Source data are provided as a Source Data file. **d** Bray-Curtis based nMDS ordination (stress = 0.11) showing differences in microbial community composition between biomes with density plot on the right showing the distribution of MDS2 values in coral, small (0.2–3 μm) and large (3–20 μm) plankton size fractions, and fish gut and mucus. **e** Prevalence and

relative abundance of ASVs in plankton, coral, and fish samples. Endozoicomonadaceae ASVs (putative symbionts) are coloured in black, Vibrionaceae (putative pathogens) in grey and all other annotations in white. I01: Islas de las Perlas, I02: Coiba, I03: Malpelo, I04: Rapa Nui, I05: Ducie Island, I06: Gambier, I07: Moorea, I08: Cook Islands, I09: Niue, I10: Upolu, I11: Wallis and Futuna, I12: Tuvalu, I13: Kiribati, I14: Chuuk Island, I15: Guam, I16: Ogasawara Islands, I17: Sesoko Island, I18: Fiji Islands, I19: Great Barrier Reef, I20: Chesterfield, I21: New Caledonia, I22: Solomon Islands, I23: Normanby Island, I24: New Britain Island, I25: Southwest Palau Islands, I26: Babeldaob, I27: Crescent Island, I28: Taiwan, I29: Oahu Island, I30: Gulf of California, I31: Clipperton Island, I32: Islas Secas.

otherwise, there was no clear pattern; some distant islands had similar communities (e.g., I4 and I27), while geographically closer islands had different communities (e.g., I12 and I18) (Fig. 5b). For *Pocillopora*, there was no clear biogeographical pattern. There were differences within the Eastern Tropical Pacific and some assemblages were similar throughout the Pacific Ocean (e.g., I05 and I17, or I06 and I16, Fig. 5c). For all coral morphotypes, variation in community composition was observed within sites and between sites of the same island (Supplementary Fig. 6).

To investigate the putative factors explaining microbial community composition in corals, we tested the effect of geographic distance and a suite of 6 environmental factors that were measured during the two years of the expedition (temperature, salinity, pH, chlorophyll a, phosphate and silicate concentrations). For *Millepora*, *Porites*, and *Pocillopora* both geographic distance and environment were significantly correlated to microbial community composition (Fig. 4d, partial Mantel test, $p < 0.01$). However, geographic distances always showed 2-3 times higher correlation values than environmental distances for all corals (Fig. 4d). Pocillopora had the highest correlation values for both geography and environmental distances. The proportion of variance explained by the environmental variables overall was: 12%, 18%, and 23% for *Millepora*, *Porites*, and *Pocillopora* respectively. We then identified the subset of environmental variables that maximized correlation with community composition, and then tested them individually with a Mantel test. For *Millepora*, salinity and seawater temperature maximised correlations with community composition ($r = 0.21$, Mantel, $p < 0.01$). For *Porites*, it was salinity and temperature ($r = 0.26$, Mantel, $p < 0.01$)), and for *Pocillopora*, salinity, $SiOH_4$, and $PO_4^{3-}$ concentrations were significant ($r = 0.19$, Mantel, $p < 0.01$).

To further verify whether microbial communities of the different coral morphotypes were structured by different factors, we assessed whether there was a relationship between the pairwise dissimilarity between microbial communities of one coral morphotype against another coral morphotype (Supplementary Fig. 8). The hypothesis was that one should observe a linear relationship if the same site-specific factors were structuring the community composition of the different coral morphotypes. The pairwise comparison between coral morphotypes showed no such significant relationship (Supplementary Fig. 8; assumption for a linear model not met). This suggests that there was no connection between similarity in community composition between sites for one coral morphotype versus another.

For all coral morphotypes, the proportion of the different taxonomic groups varied markedly between sampling sites (Supplementary Fig. 9). At the order level, *Millepora* was characterised by Kiloniellales, Spirochaetales and Cellvibrionales that were absent in the 2 other coral morphotypes (Figs. 2 and 5d). *Porites* was the only coral with Chlorobiales and *Pocillopora* had a majority of Oceanospirillales and the specific occurrence of Myxococcales and Phormidesmiales.

We used an analysis of variance partitioning to identify the putative environmental drivers of ASV abundance across samples. For each coral morphotype, we identified the ASVs that had most of their variance explained by the island or environmental factors (residual <50%, Supplementary Fig. 10). For *Millepora*, a total of 22 ASVs had >50% of

their variance explained by environmental factors (temperature, salinity, pH, chl, $PO_4^{3-}$, $SiOH_4$) or the island. The factor island was always the most important and the identified ASVs were often present at only one island (Supplementary Fig. 10). This was the case for Halieaceae (asv0000377), that was abundant on I09 only, and for a Gammaproteobacteria (asv0000157) that was exclusive to islands I07, I08 and I09. Yet other ASVs had higher prevalence and were detected on all islands, e.g., a Phycisphaeraceae (asv0003714). For *Porites*, 10 ASVs were identified, most of which were annotated as Endozoicomonadaceae. The factor island was again important, but salinity was also a strong explanatory factor for 2 Endozoicomonadaceae ASVs (asv0003573 and asv0000279) (Supplementary Fig. 10b). For *Pocillopora*, 21 ASVs were identified, most of which were also Endozoicomonadaceae. Among the other bacterial families, there were many Flavobacteria present mostly in the Eastern Tropical Pacific (I01, I02, and I32; Supplementary Fig. 10c).

The extreme size of the *Tara* Pacific dataset enables testing for the presence of shared microorganisms among samples of a given coral morphotype to assess the presence of a core microbiome (here defined as those ASVs that are consistently associated with a given coral host morphotype). *Millepora* had the highest number of prevalent ASVs, followed by *Porites* and then *Pocillopora* (Supplementary Fig. 11a). We did not detect any ASVs that were present in all coral samples. The ASV with the highest prevalence was a Vibrionales (class Gammaproteobacteria, asv0000004) present in 94% of the coral samples but with relatively low abundance (0.44% of the sequences) (Supplementary Data 1). Other prevalent ASVs in the 3 coral morphotypes included 2 Cytophagales (class Bacteroidia, asv0000019 and asv0000031) and a Rhodobacterales (class Alphaproteobacteria, asv0000077) present in 90%, 86%, and 86% of the samples, respectively. The Cytophagales sequences were 100% similar to bacteria detected previously on corals, while the sequences of Vibrionales and Rhodobacterales were 100% similar to microorganisms found in corals and a number of different marine environments. The most abundant Endozoicomonadaceae (11% of the sequences, class Gammaproteobacteria, asv0000001) was present in 80% of the samples across the 3 coral morphotypes.

In *Millepora*, the most common bacteria were a Cytophagales (asv0000019) and a Kiloniellales (asv0000126) both found in 97% of the samples (Supplementary Fig. 11b). In *Porites*, the most prevalent bacteria were a Vibrionales (asv0000004) and a Rhodobacterales (asv0000077) found in 96 and 91% of the samples, respectively (Supplementary Fig. 11c). In *Pocillopora*, a Cytophagales (asv0000019) and a Vibrionales (asv0000004) were the most common with a prevalence of 90 and 87%, followed by an Endozoicomonadaceae (asv0000003) found in 84% of the samples (Supplementary Fig. 11d) (Supplementary Data 1). For all morphotypes, the largest number of ASVs were annotated as Endozoicomonadaceae (Supplementary Fig. 11).

In terms of community composition, free-living planktonic microbial communities (size 0.2–3 μm) sampled in *Pocillopora* colony surrounding water (CSW) were significantly different from the host associated microbial communities (Fig. 2). We also demonstrated that there was no significant relationship between similarity among

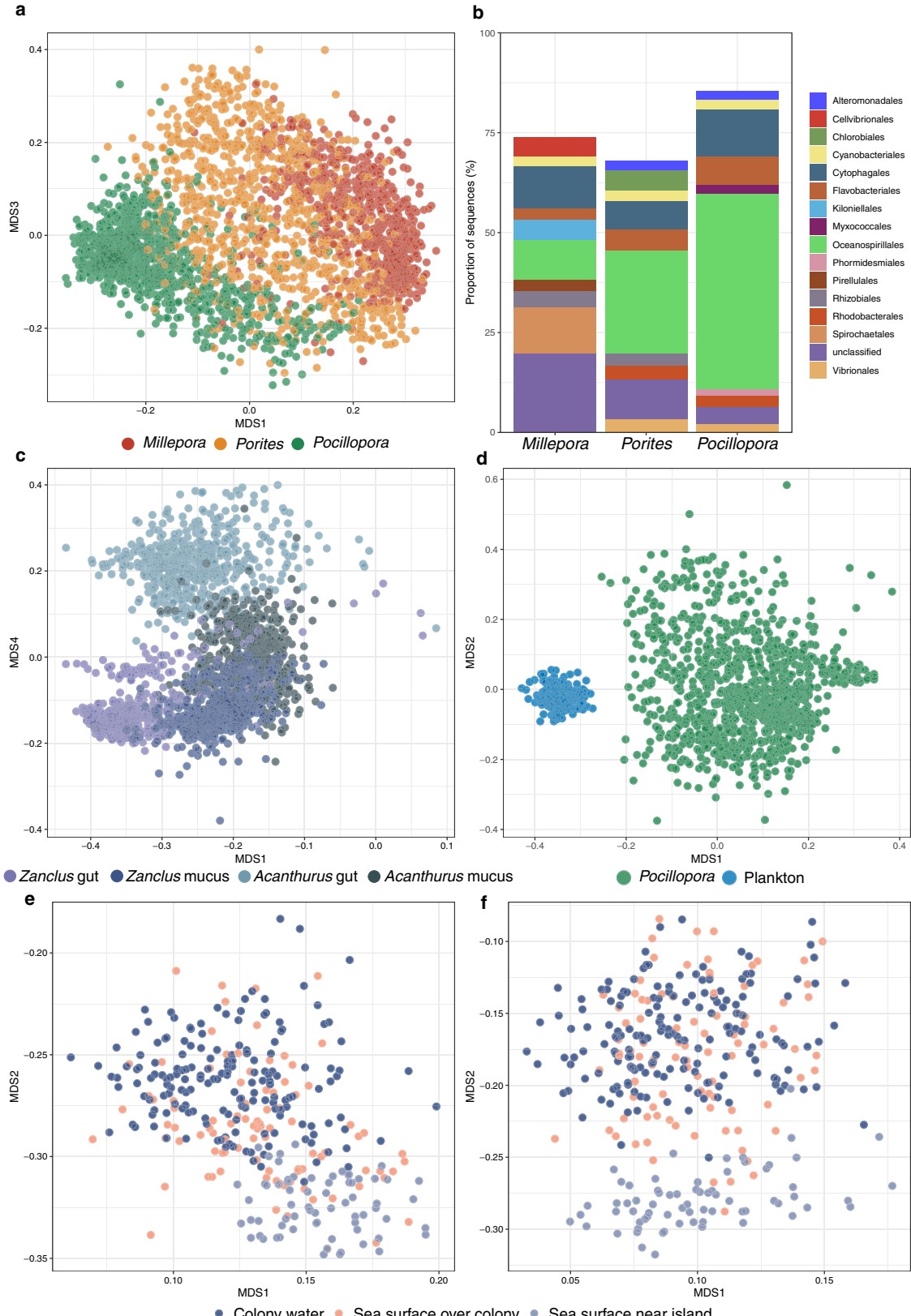

**Fig. 2 | Bray-Curtis based MDS ordinations showing differences in microbial community composition within each biomes. a** Between *Millepora*, *Porites* and *Pocillopora* and **b** their overall community composition for the 10 most abundant bacterial orders. **c** Between *Zanclus cornutus* and *Acanthurus triostegus* gut and mucus. **d** Between *Pocillopora* microbial communities and free-living planktonic communities (size 0.2–3 μm) sampled close to the *Pocillopora* colonies (colony water). **e** Between planktonic communities sampled from sea surface water near the islands, surface water over the colonies, and close to the colonies (colony water) for the 0.2–3 μm size fraction and **f** for the 3–20 μm size fraction.

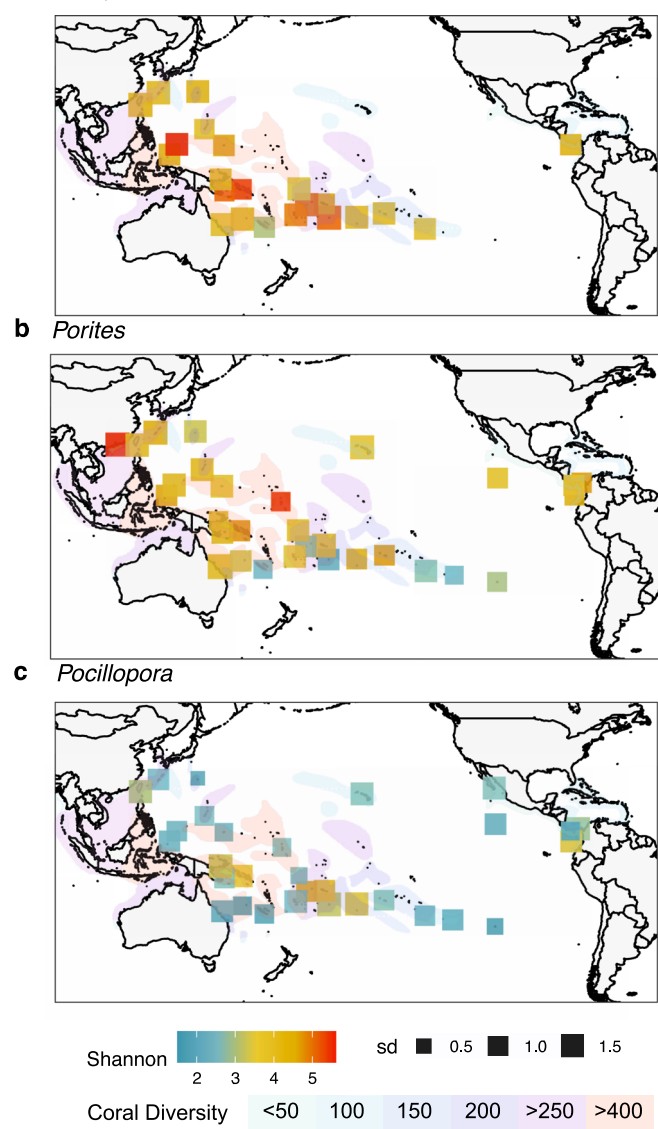

**Fig. 3 | Diversity of coral microbiomes across the Pacific Ocean.** Shannon diversity index for microbial communities of the corals **a** *Millepora*, **b** *Porites* and **c** *Pocillopora* averaged per island. Square colours represent the Shannon index strength; size indicates the standard deviation from the mean. Map colour overlay shows the diversity of the coral communities.

*Pocillopora* microbiomes and similarity among planktonic communities sampled through colonies (Supplementary Fig. 12). Finally, the microbial community composition of the planktonic communities was more homogenous within islands when compared to coral microbial communities (Supplementary Fig. 13 and 5).

Planktonic communities were more similar within sites than within islands and among islands, similar to the pattern observed for coral microbiomes (Supplementary Fig. 14). Compared to coral communities, the environment explained a much higher proportion of the variance in the planktonic communities. Environmental factors explained 37% of the variance for oceanic waters samples taken close to the islands (SRFa), and 30 and 31% of the variance for plankton at the surface over the reefs (SRF) and colony surrounding water (CSW).

## Discussion

The systematic cross-ocean sampling of the *Tara* Pacific expedition covered an unprecedented number of reefs and provides a unique view of the reef microbial diversity across coral, fish, and plankton. The associated massive sequencing effort revealed the presence of more than half a million (542,635) bacterial and archaeal 16 S rDNA amplicon sequence variants (ASVs). This represents up to 20% of the currently estimated diversity of Earth's bacterial and archaeal communities, which could encompass about 2.72–5.44 million prokaryotic ASVs[21]. Our focused, systematic and exceptional sequencing effort (2.87 billion sequences) gives a more precise insight into the diversity of the reef microbiome in comparison to previous estimates[14,18,19], and it allowed us to uncover most of the microbial diversity of the targeted biomes (98%). There were only a few ASVs shared between biomes, and our accumulation curve shows that each addition of a new biome to the sampling effort increased diversity. However, within biomes, adding animal host genera did not dramatically affect the discovery rate. For instance, based on our data, adding a third morphotype of coral increased diversity by only 3% and adding a second fish genus increased diversity by only 1.5%. Nevertheless, since there are more than 3,700 reef fish species in the Western and Central Pacific[34], and more than 600 coral species[35], a projected increase of microbial richness of even as low as 0.1% per added coral and fish species would rise the total reef microbiome diversity to 2.8 million ASVs. This number, extrapolated from a small fraction of the reef animal biodiversity (coral and fish only), is within the range of the current estimated total prokaryotic diversity for the entire Earth[21], which suggests that the global microbial biodiversity is largely underestimated. These microbes represent a huge reservoir of undiscovered and potentially important taxonomic and metabolic diversity that warrants not only future investigations, but also conservation efforts.

The *Tara* Pacific data showed that all biomes had distinct microbial communities and that the diversity of communities also differed. The plankton communities had the highest diversity compared to corals and fishes, which contradicts earlier reports showing higher diversity in corals, fish, or invertebrate surfaces compared to water microbial communities at one site[14,19]. It is, however, in accordance with a multi-reef study in Australia[18], and supports the view of a large planktonic microbiome diversity globally[26]. In our study, we targeted different size fractions of plankton that included free-living microorganisms as well as microorganisms attached to particles or associated to phyto- and zooplankton. The large microbial planktonic diversity thus reflects both the large number of microbial cells found in seawater[36], and the presence of a great number of niches within planktonic eukaryotic hosts and marine particles. Our data thus corroborates, at a large scale, that the microbiomes associated with distinct marine animal hosts have a lower diversity than planktonic microbial communities.

Different coral morphotypes had microbial diversity maxima in different islands, which implies that no common rule of diversity distribution could be defined for the 3 coral morphotypes. In particular, the coral microbiomes did not follow the general west – east gradient of diversity seen in corals[37]. The samples taken from the coral triangle (I22 to I25) did not show higher microbial diversity for *Pocillopora* and *Porites*, and the microbiome from the Eastern Tropical Pacific did not show the lowest diversity. In addition, there was no significant correlation between diversity and seawater temperature; however, the effect of heat stress, or stress generally, on microbial community diversity seen experimentally[38–41] could not be measured with our environmental sampling. Generally, it is thought that stressed colonies harbour more diverse microbial assemblages[39], likely due to stochastic processes leading to unstable microbial communities[42].

The composition of the microbiomes for the 3 coral morphotypes showed different patterns of biogeography. For each coral microbiome, these patterns did not reflect transitions between well-defined homogeneous geographical regions, but instead showed some abrupt differences between nearby islands, and similar communities between distant islands. The microbiome patterns were different from the

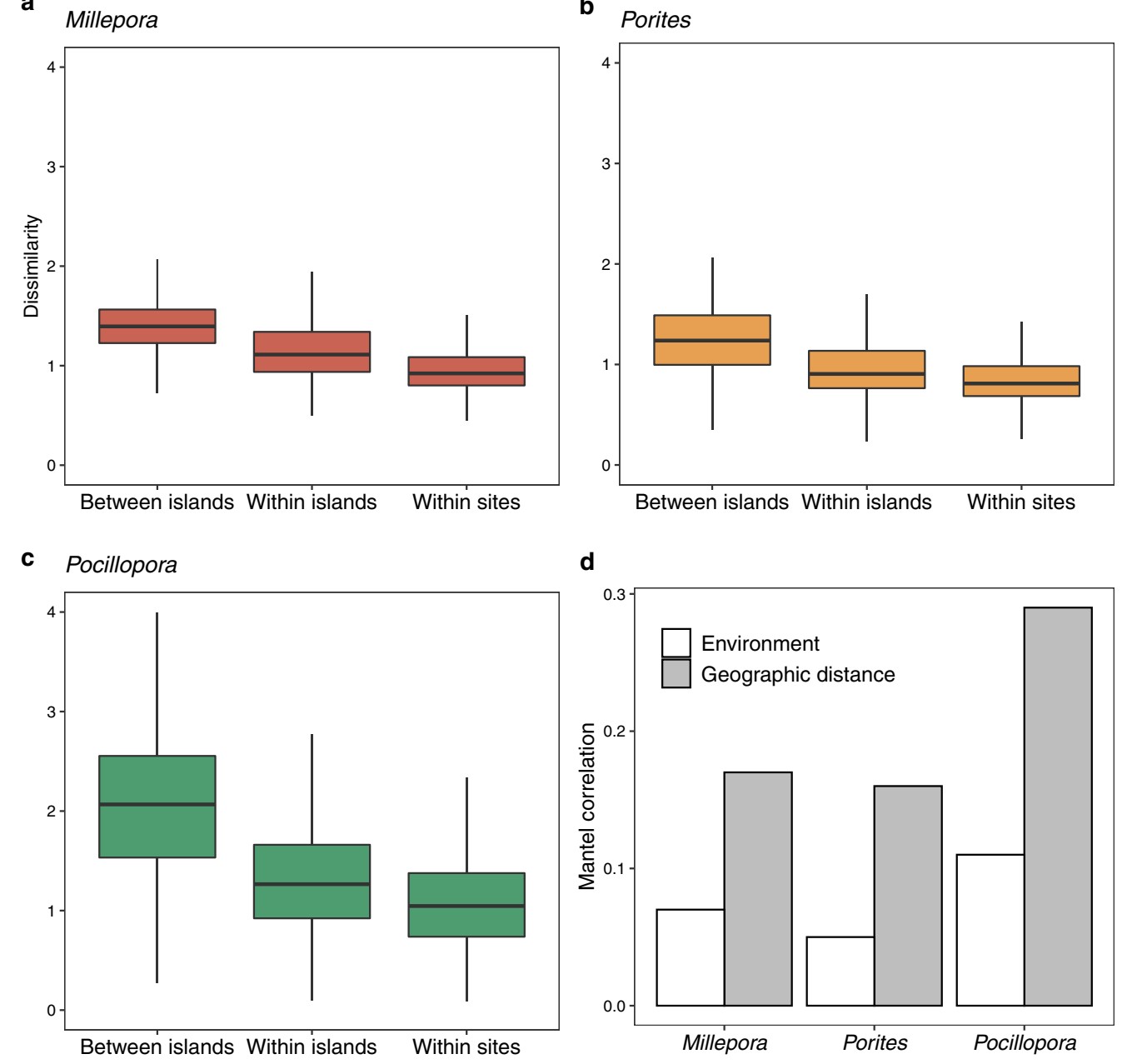

**Fig. 4 | Factors explaining the composition of coral microbiomes across the Pacific Ocean.** Pairwise dissimilarity of microbial communities compared between islands, within islands and within sites at each island for the corals **a** *Millepora*, n = 382,542 comparisons, **b** *Porites*, n = 892,080 comparisons and **c** *Pocillopora*, n = 951,600 comparisons. The dissimilarity is based on Euclidean distance computed from centred log ratio (clr) transformed ASV data. The box plot horizontal bars show the median value, the box indicates the first and third QRs, and the whiskers indicate 1.5*IQR. **d** Mantel correlation between microbial community composition and geographic distances and environmental factors for the three coral morphotypes. Source data are provided as a Source Data file.

biogeography of the coral hosts known to be separated in larger regions including Polynesia, Australia, Tonga-Samoa, Fidji-Caroline Islands, Indonesia, and Japan-Vietnam[29,43], and their sub-regions[37]. None of the 3 coral microbiomes followed the biogeographical structure of the coral hosts. On the contrary, our data show that within these regions, coral microbiomes are highly variable. Even in the Eastern Tropical Pacific, known to have coral communities homogeneous in community composition[37], there were differences in microbial communities between islands for *Pocillopora*. Our results thus demonstrate that coral microbiome composition is not driven by the same structuring factors as coral communities, thought to have a biogeography shaped by historical processes, habitat heterogeneity and species colonization ability[29]. Contemporary elements, such as host

physiological state (i.e., health, mucus production, etc.) and environmental conditions[41,44–50], are apparently stronger controlling factors of the coral microbiome than historical factors.

The variance in the coral microbiome was significantly correlated to geographical distance, and accordingly communities within-sites were always more similar to each other than to communities within-islands. This suggest that local site-specific environmental conditions drive the colonies' microbiome composition. However, the water's physico-chemical properties explained a smaller percentage of the variance of the coral microbiomes (4–11%) compared to the higher proportion of variance explained for planktonic microbial communities sampled from the water surrounding the colonies (29–30%). Locally, hosts could together be impacted by external factors that

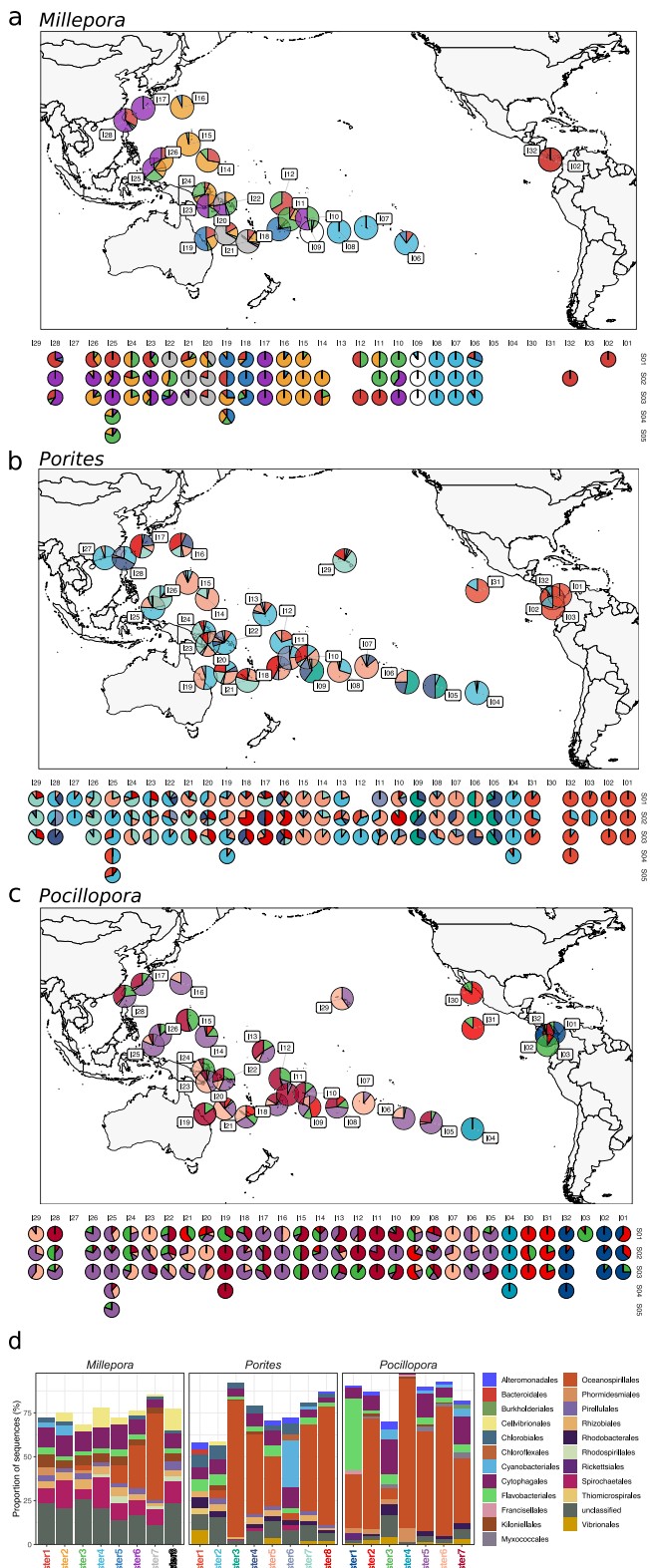

**Fig. 5 | Composition of coral microbiomes across the Pacific Ocean.** Microbial community composition for the corals **a** *Millepora*, **b** *Porites* and **c** *Pocillopora*. The pie charts represent the proportion of the different community clusters identified by hierarchical clustering. Similar colours within a figure panel represent similar microbial communities. The upper panel (map) shows summary data for each island and the lower panel represents the detail for each site at each island. An average of 10 different colonies were sampled at each site for each coral morphotype.
**d** Taxonomic composition at the order level of the community clusters. Cluster name colours correspond to the colours of the community clusters in the pie charts. Source data are provided as a Source Data file.

influence reproduction or mucus production, or environmental perturbations, such as past temperature rise or pollution, that we did not uncover with our physico-chemical measurements.

Interestingly, the absence of co-variation in community composition observed between coral morphotypes indicates that the microbiomes of the hosts have different structuring factors, which might also be related to their intrinsic propensity to flexibly host different bacteria[51]. The fact that morphotype differences in microbial community composition were large, and that the explanatory power of site was relatively higher than the explanatory power of the environmental conditions, suggests that the host itself may be the main structuring factor. A recent study on clonal *Millepora* colonies showed that microbial communities that differed between host genotype, within habitats, were potentially functionally redundant[52]. Observed differences in microbial communities may thus mask similar functional profiles of functionally redundant microbiomes[53].

Despite the observation of higher local versus global homogeneity in host specific community composition, sometimes coral microbiomes also varied within sites. Coral microbiome variability has been observed previously in different species[54–56], but results from our systematic cross ocean study enriches the existing knowledge by showing that variability is strongly species-specific[45,50]; *Millepora* microbiomes were much more stable compared to *Porites* and *Pocillopora*, the latter being the most variable. Variability has been proposed to be associated with host genotype[52], mode of reproduction[45], the age of the colony[57], or stress[50,58]. On small geographic scales, these explanatory factors are more likely than geographical or environmental factors[46,55,59–61] that cannot explain intra-site variability. It should be noted, however, that our study was conducted at the morphotype level, and we cannot exclude that some variations could be due to the presence of different species within a same morphotype.

The microbiome variability is also illustrated by the fact that there were no ASVs common to all samples within each coral morphotype[51]. The most prevalent bacteria were often *Vibrionaceae*, but since they cannot be easily distinguished based on 16 S rDNA sequences[62], common Vibrio ASVs may represent different strains. The absence of ubiquitous bacteria in this study does not challenge earlier findings reporting the presence of core coral microbiomes[19,54,63]. The concept of core microbiome depends very much on its definition which varies in terms of prevalence thresholds for common bacteria and OTU/ASV similarity cut-off[64].

Finally, the comparison of the diversity may have been influenced by the different sample preservation and extraction methods used for the different biomes (see the material and methods section). The number of bacterial cells also probably differed between samples due to differences in sample material (volume of water vs coral material vs fish skin and gut). However, the fact that the rarefaction curve was close to saturation for all biomes, and for all sample types within biomes, indicates a good coverage of the microbial diversity. It suggests that possible methodological biases were limited when doing inter-biome comparisons. It should also be noted that these comparisons targeted mainly bacteria (and not archaea) because of the use of bacterial specific primers in the first PCR step.

The unique ocean scale systematic sampling of coral reefs by the *Tara* Pacific expedition revealed an unexpectedly high diversity of microorganisms associated with plankton, coral, and fish. The fact that more than 500,000 ASVs were detected here, which represent a huge genetic diversity, suggests that the global number of microorganisms is much larger than previously thought. For corals, the ocean scale patterns of microbial diversity differed among *Millepora*, *Porites*, and *Pocillopora*, but the driver of the microbial community composition was always determined by geographic distance first, both at island and cross ocean scales, and by the environment. Our unprecedented sampling effort of the Pacific Ocean demonstrates that Earth's microbial diversity is drastically underestimated and provides new insight

into the global diversity of the coral reef ecosystem and the factors driving the distribution of its microbial world.

## Methods

### Sample collection

Samples from plankton, fish, and corals were taken in 99 different reefs from 32 island systems (noted I01 to I32) across the Pacific Ocean during the *Tara* Pacific expedition (2016–2018). Sampling protocols are detailed in Lombard et al. (2022)[65], which includes details of the coral samples used in the present study (SCUBA-3×10, protocols CTAX and CS4L in particular). Three different coral species were targeted based on morphology, *Millepora platyphylla*, *Porites lobata* and *Pocillopora meandrina*, two fish species, *Acanthurus triostegus* and *Zanclus cornutus*, and plankton in the size fraction 0.2–3 μm, 3–20 μm and 20–2000 μm, which corresponds to classical plankton size fractions separating free- living, and particle attached or eukaryotic associated bacteria. Since different coral species within the same morphotype may be difficult to discriminate by eye, we chose to present our results at the morphotype level: morphotype *M. platyphylla*, morphotype *P. lobata*, and morphotype *P. meandrina*. For the ease of comprehension, the morphotypes are referred respectively as *Millepora*, *Porites*, and *Pocillopora* in the paper. A 18 S rRNA gene based coral host analysis was also completed to identify colonies that differed the most from the most common ones in a given morphotype[66], and outlying samples were removed. For corals (sampling event *[SCUBA-3×10]*[65]), at least 3 different sites were visited at each island, and samples from 10 different individuals from each species were collected at each site using hammer and chisel, and stored individually underwater in Ziploc bags. Once on board, samples were conditioned in tubes with DNA/RNA shield (Zymo Research, Irvine, CA, USA) and conserved at −20 °C until analysis. For fish, 10 to 15 individuals of each species were sampled by spear-fishing. On board, fish mucus was sampled with a cotton swab and the digestive tract was dissected before preservation in DNA/RNA shield at −20 °C. For plankton, water samples were taken from 3 different locations: (i) as close as possible to the surface of the *Pocillopora* colonies with a diver-held hose, (ii) over the sampled coral colonies at 2 m below sea surface with a pump, and (iii) at the surface outside the reef near the islands with a pump. A total of 50 L of water was taken at each location before being filtered sequentially through 3 and 0.2 μm filters[65]. In addition, plankton larger than 20 μm were sampled at 2 m below sea surface with bongo plankton nets before being prefiltered on a 2000 μm sieve and concentrated on 20 μm filters. All filters were preserved in cryovials in liquid nitrogen.

### Environmental parameters

Environmental parameters were collected following the protocols detailed in Lombard et al. 2022[65]. Briefly, temperature and salinity were measured at each sampling site with a CTD (Castaway CTD). Water for nutrient quantification was sampled over the coral sampling sites at 2 m depth with a 5 L Niskin bottle. Two 20 mL polyethylene vials were filled running the sampled water through 0.45 μm-pore size cellulose acetate membranes. Nutrients ($NO_3^-$, $PO_4^{3-}$, $SiOH_4$) were quantified back in the laboratory from samples that were stored at −20. $NO_3^-$ was not used in the present study because of to many missing values. For chlorophyll a (chla) concentration, 2 L of water was filtered on 25 mm-diameter, 0.7-μm-pore glass fiber filters (Whatman GF/F) and immediately stored in liquid nitrogen for later High Performance Liquid Chromatography (HPLC) analysis. Sample provenance and environmental context are available on Zenodo[67].

### DNA extraction and sequencing

The different nucleic acid extraction strategies that depend on sample type are presented in detail in Belser et al.[68] (sample protocol CS4L). For fish and coral samples, which consisted of ca. 4 g of corals, 1 cm² of fish skin, and 3 cm long fish digestive tract, cells were first disrupted by

bead beating with Lysing Matrix A beads (MP Biomedicals, Santa Ana, CA, USA) on a FastPrep-24 5 G Instrument (MP Biomedicals, Santa Ana, CA, USA), and for filters (water samples), cells were disrupted by cryogenic grinding. Coral and fish DNA extractions were conducted with the commercial Quick-DNA/RNA Kit (Zymo Research, Irvine, CA, USA), supplemented for corals with an enzymatic digestion step with lysozyme, mutanolysine and lysostaphine in order to achieve an optimal lysis of the prokaryotic component of the microbiome. For water samples, extractions were done with the NucleoSpin RNA kit (Macherey-Nagel, Düren, Germany) combined with the DNA Elution buffer kit (Macherey-Nagel, Düren, Germany). DNA was quantified by fluorimetry using a Qubit 2.0 Fluorometer instrument with the Qubit dsDNA BR (Broad range) and HS (High sensitivity) Assays (Thermo-Fisher Scientific, Waltham, MA, USA). Samples were used for PCR when concentration was >1 ng/μl. Otherwise, a second DNA purification was attempted on a replicate of planktonic sample, or other homogenized suspension aliquots for coral and fish samples.

The 16 S rDNA gene was amplified by PCR with the universal 515F-Y/926 R primers[69]. Since these primers also amplify eukaryotic 18 S rRNA genes, for coral and fish, a nested PCR approach was applied with first a full-length amplification (20 cycles) using the 27 F/1492 R 16 S bacteria primer set[70,71] in order to increase the target prokaryotic DNA, and a second amplification (25 cycles) using the 515F-Y/926 Rprimers. The detailed PCR protocols are presented in Belser et al.[68]. Plankton samples were directly amplified with the 515F-Y/926 R primers. PCR amplification was performed in triplicate with the enzyme from the QIAGEN Multiplex PCR Kit (Qiagen, Hilden, Germany). A negative control was included in each PCR experiment, as well as a positive control specific to the targeted gene marker. PCR products were quantified with a Fluoroskan instrument and validated using a high-throughput microfluidic capillary electrophoresis LabChip GX system (Perkin Elmer, Waltham, MA, USA). PCR products were pooled after amplification and cleaned using AMPure XP beads. Libraries were prepared using the NEBNext DNA Modules Products (New England Biolabs, MA, USA) and NextFlex DNA barcodes (BiOO Scientific Corporation, Austin, TX, USA) with 100 ng of purified PCR product as input. Libraries were subjected to Illumina sequencing at the Genoscope (Evry, France). The taxonomic assignation of negative controls sequences were used to build a database of possible contaminant bacteria DNA that can be present in reagents. The database was used to remove potential contaminant sequences from the dataset. All sequencing files were submitted to the European Nucleotide Archive (ENA) at the EMBL European Bioinformatics Institute (EMBL-EBI) under the *Tara* Pacific Umbrella project PRJEB47249. Samples and their metadata were registered in the ENA biosample database.

### Sequence analysis

An ASV abundance table was built with DADA2 v1.14 as detailed in the scripts published in Zenodo[72]. Samples were grouped by sequencing lane to learn errors and infer ASVs. Resulting ASVs from forward and reverse reads were then merged and chimeric sequences were removed using DADA2 internal functions. ASVs representing less than six inserts were tagged as being spurious and removed. Taxonomic annotation was performed with IDTAXA[73] with a confidence threshold of 40 against the SILVA v.138 database. Eukaryotic ASVs (chloroplast and mitochondria) were identified based on taxonomic annotation following the criteria published in Zenodo[72] and removed from the dataset prior to analysis. In addition, bacterial sequences annotated at the family level as Oxalobacteraceae, Comamonadaceae, Cutibacterium and Yersiniaceae were identified as reagent contaminants and removed from the dataset.

### Data analysis

The accumulation curve was constructed, after removing ASVs present in only one sample, with the "accumresult" function with 100

permutations in the R package BiodiversityR. The total extrapolated richness was estimated with the function "veiledspec" from a fitted Preston model ("prestondistr") in vegan. The model of species abundance distribution fitted the log-normal model according to the Akaike's Information Criterion (function "radfit" in vegan). The Shannon index was computed with the function "diversity" in the package diverse after removing samples with less than 50,000 sequences (remaining samples n = 5,298). Differences in diversity were tested with the unpaired Wilcoxon test ($p < 0.01$) after adjusting p values for multiple testing using Benjamini and Hochberg method (FDR) with the function "pairwise.wilcox.test" in the package stats. The nMDS ordination on a rarefied dataset was constructed with Hellinger transformed resampled data with metaMDS function in vegan. Sample size was rarefied to 15000 sequences per sample with the package rtk. Significant differences between community composition were tested with PERMANOVA with the adonis function of the vegan package. Maps were built with the function geom_map using the "world2" map retrieved with the function map_data in R.

Comparison of beta diversity between islands, between sites and within sites is based on Euclidean distances computed from weighted clr normalised counts ("CLR" function in easyCoda, package). Significant differences between groups were tested with the unpaired Wilcoxon test ($p < 0.01$) after adjusting p values for multiple testing using Benjamini and Hochberg method. Betadispersion was computed with the function " Betadisper" in vegan from Euclidian distances ("parDist" function) after clr transformation of the data.

To infer microbial communities at the island level, all microbial communities were grouped into community clusters by hierarchical clustering based on Euclidean distances. We then plotted the relative abundance of these different community clusters in each island (Supplementary Fig. 6). The composition in community clusters for each island was then used to identify similarity in microbial communities between islands by hierarchical clustering. The best number of clusters (8 for *Porites*, 8 for *Millepora* and 7 for *Pocillopora*) was determined with the "NbClust" function in the NbClust package with the ward.D method that gave the strongest clustering structure. Clustering was done with "hclust" in vegan.

Fisher's exact test was used to test if the pattern of microbial community composition across the Pacific Ocean was different between the 3 coral morphotypes. The test, which determines if there are non-random associations between categorical variables, was conducted with the function "fisher.test" in R.

The effect of environmental factors and geographic distance on community composition was tested with the partial Mantel test using the function "mantel.partial" in vegan. To test the effect of the geography, the community composition was used as the response, the geographic distance as the predictor, and the environmental distance as the condition factor. To test the effect of the environment, the roles of environmental distance and geographic distance were interchanged. The matrices of microbial community composition were computed as Euclidean distances of the clr transformed ASV abundance table. The matrices of geographic distances between sampling points were computed based on their coordinates with "distm" in the geosphere package with the haversine method to account for the curvature of the Earth. The matrices of environmental Euclidean distances were calculated with "dist" in vegan.

The proportion of variance in community composition explained by the environmental factors that we measured was calculated with a redundancy analysis (RDA) conducted with "rda" in vegan. The analysis was based on clr transformed ASV abundances in relation to sea water temperature, salinity, pH, concentrations of phosphate ($PO_4^{3-}$), silica ($SiOH_4$) and chorophyll a. The overall significance of the ordination was tested with an ANOVA. The combination of environmental variables that best explain changes in community composition was identified with the "bioenv" function

using Spearman's rank correlations in vegan. The significance of the individual parameters identified in "bioenv" were then tested individually with a mantel test.

A variance partitioning analysis was conducted to identify the individual ASVs whose variance was best explained by environmental factors. The analysis was done with the function "fitExtractVarPartModel" in the variancePartition package in R.

### Reporting summary
Further information on research design is available in the Nature Portfolio Reporting Summary linked to this article.

## Data availability
Sample provenance and environmental context are available on Zenodo[67]. The ASV abundance table is available on Zenodo[72]. Samples and their metadata were registered in the ENA biosample database. All sequencing files were submitted to the European Nucleotide Archive (ENA) at the EMBL European Bioinformatics Institute (EMBL-EBI) under the Tara Pacific Umbrella BioProject PRJEB47249. All other data supporting the findings of this study are provided in Supplementary Information or Source Data file. Source data are provided in this paper.

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

## Acknowledgements

Special thanks to the *Tara* Ocean Foundation, the R/V Tara crew and the *Tara* Pacific Expedition Participants (https://doi.org/10.5281/zenodo.3777760). We are keen to thank the commitment of the following institutions for their financial and scientific support that made this unique *Tara* Pacific Expedition possible: CNRS, PSL, CSM, EPHE, Genoscope, CEA, Inserm, Université Côte d'Azur, ANR, agnès b., UNESCO-IOC, the Veolia Foundation, the Prince Albert II de Monaco Foundation, Région Bretagne, Billerudkorsnas, AmerisourceBergen Company, Lorient Agglomération, Oceans by Disney, L'Oréal, Biotherm, France Collectivités, Fonds Français pour l'Environnement Mondial (FFEM), Etienne Bourgois, and the Tara Ocean Foundation teams. This study was supported in part by FRANCE GENOMIQUE (ANR-10-INBS-09). S.S. acknowledges the support of the Swiss National Science Foundation grant 205321_184955. *Tara* Pacific would not exist without the continuous support of the participating institutes. The authors also particularly thank Serge Planes, Denis Allemand, and the *Tara* Pacific consortium. This is publication number #23 of the *Tara* Pacific Consortium. Sampling permits are provided as supplementary material.

## Author contributions

P.E.G. conceptualized and designed the study, analyzed the data, and wrote the manuscript; H.J.R. and G.S. conducted the bioinformatic analyses of the raw sequence data; C.H. and N.H. assisted in data analysis. B.C.C.H., D.A.P.G., M.Z., C.R.V., R.V.T., and S.S. provided useful comments and manuscript edition. P.E.G., H.J.R., G.S., C.H., N.H., B.C.C.H., P.H.O., A.P., K.L., Ca.B., Emi.B., Sa.R., J.P., G.B., G.I., C.M., E.J.A., D.A.P.G., M.Z., S.A., B.B., Emm.B., Ch.B., Cd.V., E.D., M.F., D.F., P.F., E.G., F.L., St.P., St.R., O.P.T., R.T., D.Z., C.R.V., R.V.T., S.S., P.W., D.A., and Se.P. reviewed the manuscript.

## Competing interests

The authors declare no competing interests.

## Additional information

Pierre E. Galand ●[1,2] ✉, Hans-Joachim Ruscheweyh ●[3], Guillem Salazar ●[3], Corentin Hochart ●[1], Nicolas Henry ●[2,4], Benjamin C. C. Hume ●[5], Pedro H. Oliveira[2,6], Aude Perdereau[2,6], Karine Labadie ●[2,6], Caroline Belser ●[2,6], Emilie Boissin ●[7], Sarah Romac ●[4], Julie Poulain[2,6], Guillaume Bourdin ●[8], Guillaume Iwankow[7], Clémentine Moulin[9], Eric J. Armstrong ●[7], David A. Paz-García[10], Maren Ziegler[11], Sylvain Agostini ●[12], Bernard Banaigs[7], Emmanuel Boss ●[8],

Article

Chris Bowler ®[2,13], Colomban de Vargas[2,4], Eric Douville ®[14], Michel Flores ®[15], Didier Forcioli ®[16,17], Paola Furla ®[16,17], Eric Gilson ®[16,17,18], Fabien Lombard ®[2,19,20], Stéphane Pesant[21], Stéphanie Reynaud ®[17,22], Olivier P. Thomas[23], Romain Troublé ®[2,9], Didier Zoccola ®[17,22], Christian R. Voolstra ®[5], Rebecca Vega Thurber ®[24], Shinichi Sunagawa ®[3], Patrick Wincker ®[2,6], Denis Allemand ®[17,22] & Serge Planes[2,7]

[1]Sorbonne Université, CNRS, Laboratoire d'Ecogéochimie des Environnements Benthiques (LECOB), Observatoire Océanologique de Banyuls, Banyuls sur Mer, France. [2]Research Federation for the Study of Global Ocean Systems Ecology and Evolution, FR2022 GOSEE, Paris, France. [3]Department of Biology, Institute of Microbiology and Swiss Institute of Bioinformatics, ETH Zürich, Zürich, Switzerland. [4]Sorbonne Université, CNRS, Station Biologique de Roscoff, AD2M, UMR 7144, ECOMAP, Roscoff, France. [5]Department of Biology, University of Konstanz, Konstanz, Germany. [6]Génomique Métabolique, Genoscope, Institut François Jacob, CEA, CNRS, Univ Evry, Université Paris-Saclay, Evry, France. [7]PSL Research University: EPHE-UPVD-CNRS, UAR 3278 CRIOBE, Laboratoire d'Excellence CORAIL, Université de Perpignan, Perpignan, Cedex, France. [8]School of Marine Sciences, University of Maine, Orono, USA. [9]Fondation Tara Océan, Paris, France. [10]Centro de Investigaciones Biológicas del Noroeste (CIBNOR), La Paz, BCS, México. [11]Department of Animal Ecology & Systematics, Justus Liebig University Giessen, Giessen, Germany. [12]Shimoda Marine Research Center, University of Tsukuba, Shimoda, Japan. [13]Institut de Biologie de l'Ecole Normale Supérieure (IBENS), Ecole normale supérieure, CNRS, INSERM, Université PSL, Paris, France. [14]Laboratoire des Sciences du Climat et de l'Environnement, LSCE/IPSL, CEA-CNRS-UVSQ, Université Paris-Saclay, Gif-sur-Yvette, France. [15]Weizmann Institute of Science, Department of Earth and Planetary Sciences, Rehovot, Israel. [16]Université Côte d'Azur, CNRS, INSERM, IRCAN, Medical School, Nice, France. [17]LIA ROPSE, Laboratoire International Associé Université Côte d'Azur-Centre Scientifique de Monaco, Monaco, Principality of Monaco. [18]Department of Medical Genetics, CHU of Nice, Nice, France. [19]Sorbonne Université, Institut de la Mer de Villefranche sur mer, Laboratoire d'Océanographie de Villefranche, Villefranche-sur-Mer, France. [20]Institut Universitaire de France, Paris, France. [21]European Molecular Biology Laboratory, European Bioinformatics Institute, Wellcome Genome Campus, Hinxton, Cambridge, UK. [22]Centre Scientifique de Monaco, Monaco, Principality of Monaco. [23]School of Biological and Chemical Sciences, Ryan Institute, University of Galway, Galway, Ireland. [24]Microbiology Department, Oregon State University, Corvallis, OR, USA. ✉e-mail: pierre.galand@obs-banyuls.fr

