## [Peer Review File · Nature Communications]

Diversity of the Pacific Ocean coral reef microbiomeReviewer #1 (Remarks to the Author):

This is an important study and I congratulate the authors for organising and ultimately pulling off such a large sampling campaign. The ultimate findings about the enormous microbial diversity estimated for the Pacific, based on a few key biomes across the region need to be disseminated as widely as possible, and I have no doubt that these results will be highly impactful when published.

-We appreciate the positive comment of the reviewer.

Unfortunately, there are currently many caveats with big consequences on the quality of the paper. My major concerns are about samples identity, methods description, depth of analysis, depth of discussion, figures quality, and quality of the introduction.

Specifically:

1. Remove fish and plankton.

Why bother sampling and sequencing the gut and mucus of the fish, and 2 types of “plankton” (see below for the use of brackets), without analysing these data further than in a very, very superficial way? The title and introduction of the paper sell an analysis on “the coral reef microbiome” but the gap is so gigantic between what was done with coral versus non-coral (1 figure versus 6 figures in the main text), that you should either analyse the fish and “plankton” as they deserve to be analysed, or remove them completely and have an honest title talking about the diversity of the “coral microbiome across the Pacific”. Sure, there’s a gradient of coral diversity across the Pacific. So what? Fish diversity isn’t homogenous in the Pacific either. And even if it was, you’ve got all these fish samples and you can’t even present the taxonomy of fish gut versus fish mucus separately (fig. S5), even though you tell us that they’re different (line 169-170)? I cannot believe that these data would not be analysed thoroughly, so I’m assuming that they will be, in a later paper, maybe, but then why are you talking about them here? My advice is: focus on the coral; publish 2 other papers with fish and plankton, properly.

-We understand the reviewer’s comment because we went through similar thoughts when designing the paper. The dilemma is whether to publish everything in one article, which would be overwhelming, or publish it separately, but with the risk of missing the larger picture. We concluded that we needed all biomes (water, fish and coral) to describe the very large diversity of reef microorganisms. Restricting the diversity section to corals only would considerably reduce the impact of the results. On the other hand, for betadiversity, biogeography and community composition we could not go in-depth for all the biomes because the number of figures, results and discussion associated would go beyond the scale of a research article. We, therefore, chose for betadiversity to focus on corals only, which are the emblematic organism of the reefs, and to later publish the fish and water data in separate, focused papers. We hope that the reviewer will support our choice.

1. I don’t agree that the paper would be less impactful if you were focusing on corals only, because that’s already a massive dataset. But this is not a deal breaker and when all your data is published in depth, it is true that it will be nice to have that one overarching view. Personally, I would not have done it that way, but I can now understand your point of view.

2. There are issues with what the text says about sample types and what samples actually really are. First of all, you say that you’ve done your analysis on 3 species of corals, which are *Millepora platyphylla*, *Porites lobata* and *Pocillopora meandrina* (line 118-119). But in the methods you say that “Since different coral species within the same genus may be difficult to discriminate by eye, we chose to present our results at the genus level: *Millepora*, *Porites*, *Pocillopora*. A 18S rRNA based coral host analysis is available to identify colonies that differed the most from the common ones in a given genus (Zenodo, 2020)”. This is very confusing. It sounds like you actually did not genotype all your samples, so you do not know

for sure that all the samples you collected were from these 3 species, and you therefore decided to use genera instead of species names. So maybe your *Millepora* samples have representatives of 3 species and your *Porites* just one. You cannot do a comparison of microbial diversity when you have no idea how many host species you're looking at. All samples should have been genotyped as is done routinely in coral microbiology, precisely because the community knows that there are cryptic coral species and that the species identity is crucial for data interpretation.

-The reviewer rises an important question. Unfortunately, we could not genotype the 2540 samples of corals used in this study. We acknowledge that it's a limitation of the paper and we, therefore, chose to write about genus rather than species. However, we conducted an 18S survey, which is not as precise as genotyping, but allows cost-effective discrimination of outlying colonies within a species. All outliers within each species were removed from the analysis. With the means available, we considered that it was a good compromise. We have added a sentence to the discussion to highlight the limitation of the study (654-567).

2. OK, 2 things here:

2.1 The 18S analysis is not detailed here, and I do not have access to the Hume dataset, which could be where this analysis is (line 503, reference 68), or is it reference 34 as stated line 568? Apart from this, I believe the Lombard et al paper details your sampling. Here is where things do not add up in this paper with your response to my comment: section 2.1 of the Lombard et al. paper (lines 205-211) states that the samples collected for the analysis presented in the manuscript I am currently reviewing were SCUBA-3X10, for which photos were taken. Then, section 2.9.1 (lines 328 to 345) states that for these samples, taxonomic identification was done based on morphology (CTAX). It appears that the only 18S analysis was done for the SCUBA-SURVEY samples, which were used for coral host diversity, not for the present manuscript. Nowhere is it said that the SCUBA-3X10 were a subset of the SCUBA-SURVEY. Therefore, it looks like the samples collected for the present study were only assigned taxonomy through pictures and morphological observation, not 18S. Maybe it's all in the zenodo files but please clarify.

2.2 Best case scenario, even if you were able to identify the genus of each coral with 18S, your choice of using the term "genus" rather than "species" does not address the problem; because you are actually not analysing your data at the genus level (which makes your statement "our study was conducted at the genus level", line 465, incorrect). Using the term "genus" creates the opposite flaw as using "species", because you are now assuming that the morphologies that you collected, which were very openly and intentionally restricted to three very specific ones (with the hope of catching only three species altogether), host enough species that they are representative of 3 entire genera. The only way that using "genus" would be correct is if you had done your analysis at the genus level indeed, that is if you had sampled at the genus level, that is if you had sampled multiple species; enough species, in fact, so that collectively they would be representative of each of the genera. If you had done this, you could have said "Within genus X, we found that much diversity". But given how you sampled, as much as you cannot say that you only have one species per genus, you cannot extrapolate the microbial diversity that you see in corals showcasing the morphologies of *Millepora platyphylla*, *Porites lobata* and *Pocillopora meandrina* to the entire genera *Millepora*, *Porites* and *Pocillopora*, respectively. Granted, *Millepora* only has a couple of species, but this is not at all the case for the other two genera. That is why you are not doing your analysis at the genus level. You might be doing it at the species level, but you don't know that. You are actually doing this analysis at the "morphology level".

What is done is done, all you can do now is address the problem. To do so, I recommend this: you need to use the term “morphology” instead of “species” or “genus”. This is the only accurate way of naming what you’re looking at. I advise using something like “morph_MPla”, “morph_PLob” and “morph_PMea”, for example, to nominate your sample types. It is the only way of ensuring accuracy (and minimising future mis-citation).

3. This issue of “what are you looking at here?” is also valid for what you call “planktonic microbes”. Reading the methods, it sounds to me that you collected 2 types of samples. First type: pelagic microbes, often called “seawater microbiome” in the literature, because you filtered seawater through 3µm and 0.2µm, you kept the filters and extracted DNA from this, which is exactly what people do when they look at free-living seawater microbiomes. Second type: plankton-associated microbes, i.e microbes that associate with phytoplankton or zooplankton as large as 2mm (plankton-associated microbes). These are different types of microbes and we hear nothing from it for the entire results and discussion, and I don’t believe you should lump them in the term “plankton microbes”.

-We acknowledge that the result, discussion and method section were not precise enough regarding plankton. We now write in the method that the size fractions correspond to classical plankton size fractions separating the free living, and particle attached or eukaryotic associated prokaryotes (lines 600-601). We also now specify in the result section that: “...the small size fraction that corresponds to the free living microorganisms (<3 µm)... when adding larger plankton size fractions that include microorganisms associated to planktonic eukaryotes or particles (Fig. 1a).” lines 211-213, “Plankton communities, including free-living and attached prokaryotes, ...” lines 218-219, “For both the free living (0.2 µm - 3 µm) and the particle or eukaryote associated (3 µm - 20 µm) size fractions of plankton...” lines 230-231, “In terms of community composition, free-living planktonic microbial communities (size 0.2 – 3µm)...” line 430. We also specify the type of plankton in Supplementary Fig. 13-16. In the discussion: “In our study, we targeted different size fractions of plankton that included free-living microorganisms as well as microorganisms attached to particles or associated to phyto- and zooplankton. The large microbial planktonic diversity thus reflects both the large number of microbial cells found in seawater (Whitman et al. 1998), and the presence of a great number of niches within planktonic eukaryotic hosts and marine particles.” Line 481-487.

3. OK, that works.

4. A lot of this confusion stems from the fact that the sampling protocols are not detailed, since they will be in a later publication (line 480). This is inappropriate. One should be able to reproduce your study; not that they would, at this scale, probably, but that’s not the point. For example, what is the plankton collected “near the island”? How far were samples collected “within sites” or “within islands”? This is all very vague and never explained. If the Lombard et al. paper had been submitted to the same journal at the same time, reviewers might have had access to sampling protocols but this does not seem to be the case, as the article is “in prep”. Right now, we have no idea when the methods will be available.

-We apologise for not providing the method paper together with our submission. The paper on sampling protocols (Lombard et al) and on the omics approach (Belser et al) are now both accepted in the journal Scientific Data. Awaiting publication, the reviewers can access Lombard et al. from bioRxiv (<https://www.biorxiv.org/content/10.1101/2022.05.25.493210v1>) and Belser et al. from arXiv (<https://arxiv.org/abs/2207.02475>).

We have also extended the method section and in particular for the water sampling (lines

614-617), environmental parameters (626-630), and DNA extraction and sequencing. See answers to reviewers 1 and 3.

4. OK.

5. The discussion starts with the important findings about the huge microbial diversity uncovered, which is great. But there are a few things that are either questionable or not discussed. Generally speaking, I feel like there's a lack of "depth" in the discussion (sorry, this is quite vague to say, so probably not very constructive).

Lines 396-397: you write "Our data can thus suggest that the microbiomes associated with distinct marine hosts have a lower diversity than planktonic microbial communities" (even in this sentence, it's not clear what you mean by "planktonic microbial communities", because you oppose them to "distinct marine hosts"). I would rather say that your data corroborate previous findings, with the studies that you cite earlier in this paragraph. Basically, you're confirming on a big scale what has been found with dozens of comparisons between host and seawater microbiomes...It's a bit different than saying that your data suggest something. This isn't the big finding of your work anyway so it's OK if it's been shown before.

-We have now changed the sentence to distinguish the different planktonic compartments and specify "distinct marine animal hosts" line 488. We now write that our data corroborate previous findings in line 487.

5. OK.

Lines 398-408 is a nice description of the results but for a paper submitted to this journal, I was expecting to have these results put into a broader ecological context. For example, has anything been done on terrestrial systems about this? Is there any ecological theory that suggests that host diversity is mirrored in microbial diversity? I'm thinking about the rainforests for example: is microbial diversity higher in rainforests, which harbour greater eukaryotic diversity, than it is in other types of forests? Or even in the marine environment, at a smaller scale? It's not obvious to me that one would assume that microbial diversity follows eukaryotic diversity. These are such different types of organisms living with such different constraints. This should definitely be discussed more.

-Very good point. We haven't found any literature on the link between host diversity and microbial diversity in marine systems, but thought that it would be interesting to test the hypothesis. The idea is that a higher number of coral species would lead to a higher number of different microorganisms diffusing into the water, and potentially colonising other hosts. The soil literature is often richer than the marine one, and although we did not find studies directly on symbiont vs host diversity, some are comparing rainforest and pasture soil microbial diversity. Results are contrasted with studies showing higher diversity in forest soils (Paula et al. 2014), no differences (Mueller et al. 2014, Mirza et al. 2014), or higher in pastures (Rodrigues et al. 2012). We also found one study linking the diversity of pathogens and symbiotic fungi to plant diversity (Wang et al. 2019), and one on French soils showing large variations of diversity linked to the soil's characteristics (Terrat et al. 2017). These studies are, however, not related to our question and since we don't have any general ecological theory to frame the host vs microbiome diversity theory, we changed the paragraph to focus first on the finding that no common rule of diversity distribution could be defined for the 3 coral genera (lines 490-495).

Paula, F.S., Rodrigues, J.L.M., Zhou, J., Wu, L., Mueller, R.C., Mirza, B.S., Bohannan, B.J.M., Nüsslein, K., Deng, Y., Tiedje, J.M. and Pellizari, V.H. (2014), Land use change alters functional gene diversity, composition and abundance in Amazon forest soil microbial communities. *Mol Ecol*, 23: 2988-2999.

<https://doi.org/10.1111/mec.12786>

Mueller, R., Paula, F., Mirza, B. et al. Links between plant and fungal communities across a deforestation chronosequence in the Amazon rainforest. *ISME J* 8, 1548–1550 (2014). <https://doi.org/10.1038/ismej.2013.253>

Mirza, Babur S., et al. "Response of free-living nitrogen-fixing microorganisms to land use change in the Amazon rainforest." *Applied and Environmental Microbiology* 80.1 (2014): 281-288.
Rodrigues, Jorge LM, et al. "Conversion of the Amazon rainforest to agriculture results in biotic homogenization of soil bacterial communities." *Proceedings of the National Academy of Sciences* 110.3 (2013): 988-993.

OK.

Line 455-461: your definition of a core microbiome here is based on the presence of an ASV in 100% of samples. This is never done. Just because no study conforms to your definition of the core microbiome, which is an ASV (not even an OTU!) found in 100% of samples, does not mean that your findings “challenge the idea of a core microbiome”. Even you acknowledge that: lines 460-461 destroy line 459 so why even go there?

-We have changed the sentence accordingly (line 584). We would like to emphasise that our point here was not to define a core microbiome per se, but rather to check within this very large dataset if we could find ubiquitous microorganisms.

OK.

Line 436: you can't really say that you're talking about 3 hosts because coral microbiomes vary between species, and you don't know how many species you have.

-We have changed the sentence to avoid confusion (line 545).

6. Introduction structure.

While the introduction contains all the necessary information, it needs serious restructuring. In your 2nd paragraph, you start describing (line 82) how microbes are important for corals, and then you say why they're important for fish. And then for plankton. The truth is, they're important for octopuses as well, and probably sharks too, not to mention anemones. The way it's written feels quite contrived and it sounds like you try to justify why you focused on these groups rather than others, when you probably simply had to make a choice and could not sequence 200 types of organisms for logistics and financial reasons. I'm sure people will understand that considering your sampling effort, you had to make choices, and most readers of this journal now know that microbes are important for all organisms.

There is a way of keeping the “funnel principle” in the introduction, which would work better, where you go from a wide concept at the beginning to a narrow one at the end. To ease the flow of the introduction, I strongly suggest the following: after paragraph 1, keep the sentence about “Microbes are the invisible yet essential...role of symbiosis”. After this sentence, put your paragraph about the rationale (the one starting on line 100 “Despite the fact that corals represent hotspots of diversity” and finishing with “at the ocean scale remains unknown”), and then you say something like “To address this question, we have designed a study to assess the diversity of the Pacific ocean reef microbiome. We chose to focus our efforts on 3 types of organisms, which fulfill crucial ecological roles on coral reefs: corals, fish and plankton (ref, ref, ref)”. Then you say something like “Because strong patterns of diversity exist for the coral hosts across the Pacific with higher densities close to the coral triangle (ref, ref, ref), we further investigated patterns of diversity for the corals microbiome across the Pacific basin”. That would work better. Obviously, that's if you analyse fish and plankton as thoroughly as corals. And that's also if by “plankton” you mean “plankton-associated microbes”; otherwise it's just seawater, in which case you need to rephrase, etc...If you opt for what I think is a wiser choice (removing non-coral, see above), your intro will change anyway. But who knows, this might be a useful comment for your future version regardless.

-We appreciate the constructive comment of the reviewer. We have modified the introduction following his/her precise recommendations. However, we chose to have the “question” section at the end of the introduction to avoid repeating the goals of the study.

OK, that works !

6. Figures, typos, semantics

Overall comment to all your responses for figures, typos and semantics so far:

Nice work, the figures look much better now, you've improved readability and it's much more pleasant to the eye. Just a note here: I concur with you about keeping the numbers on the world map in the order of sampling. Many of your readers do not usually read from left to right, anyway...

New comments about grammar:

Line 78: I would say "of marine ecosystems" (plural rather than singular)

Line 82: change "diversity of living species" to "species diversity"

Line 85: it's a small thing, but if I read "microorganisms" after reading a paragraph on corals, I think "Symbiodinaceae". It's a bit strange to me that there is no mention of it in the introduction. But I understand that you did not analyse this here so it's fine if you'd rather ignore them altogether. Just a personal comment I guess.

Line 92: replace "as it was often assessed" by "as it has often been assessed"

Line 94: replace "recent attempts of counting" by "recent attempts at counting".

Line 108: remove "also" (because you already have "moreover").

Line 186: "The most abundant ASV belonged to the family Endozoicomonadaceae...". Since you're talking about abundance here, please tell us how abundant it was (even if you just give max and min across all morphologies).

Line 382: change "our accumulation curve suggests" with "our accumulation curve shows", because there is no ambiguity in this result.

Supplementary figures are important, too, and it feels like these have been neglected. Not the best look, especially for a journal like this one.

-We have carefully checked all supplementary figures.

Fig 1a does not do justice to the amount of work that this represents. What makes this study unique is precisely the sampling design. The map should not be stretched like this, but similar to the one in Fig 4.

-We have changed the map.

For Fig. 4b, maybe give examples in the text about which islands were distant but similar and which were close but different (I4-I27 for "distant but similar" and I12-I18 for "close but different"?).

-We now give examples (lines 321-322).

Fig. S5: definitely do not cluster fish gut and mucus together. Also, small comment but put the Cyanobacteria in a dark grey or something because this light pink next to the orange is really hard to see.

-We now separate gut and mucus and changed the colours of the legend.

Fig. S9: "separating to samples" is a typo.

-Corrected.

Fig. S10: the text here is way too small. From what I can see, it's all the same legend for the 3 figures so reduce the size of each figure so that you can fit all 3 on a single page and have one, bigger legend that people can read.

-We have increased the font size, grouped the figures and provided a single legend for all 3. The quality of the figure has greatly improved.

Fig. S15 appears cut in my document.

-Corrected.

Line 371 says 2.87 billion sequences. But line 146 it's 2.84 billion sequences. Which is it?

-We apologise for the error. It's now corrected, it was 2.87.

Line 396 and line 428: remove "can" and "could", your data "suggest".

-The first sentence was changed following reviewer1's comment: "Our data thus

corroborates, at a large scale, that the microbiomes” (line 487) and could was removed from the other sentence.

Line 487: “was” should be “were”.

-Corrected.

Line 465: microorganisms do not associate with water, they live in it. But they associate with coral, fish and plankton. See my earlier comment on plankton definition.

-We now write plankton instead of water.

Reviewer #2 (Remarks to the Author):

This is an unprecedented dataset of coral reef organismal and seawater biomes that provides novel and surprising insights into the bacterial diversity of these biomes. I am supportive of the presentation of the data and findings, and find the paper well written.

I'm surprised that plankton microbial diversity was higher than the animal-based diversity. This contrasts with my view of these systems. Although, now I realize that many studies do not directly compare richness/diversity comparisons between seawater and coral and instead focus on beta diversity and compositional parameters. I don't have an ask for the reviewers here, just a comment.

There was just one thing that I think the authors could improve upon, which would strengthen the acceptance of this paper's findings by the larger microbiology community. The paper presents non-identical methods for sample storage (deep frozen vs. preservation fluid) and processing (different DNA extraction methods), yet ultimately compares all data similarly. I understand that the authors had a trade off in terms of sample preservation and processing strategies. Additionally, some sample types (like the seawater analysis) seemed to be tied to methods from prior global analyses, and thus different preservation and DNA extraction methodologies were used. To acknowledge this potential bias in the data, the authors should include a short caveat section that addresses the methodological limitations directly. Additionally, this section should address the potential microbial abundance/biomass differences between the seawater (50 L of water!) and coral tissue pieces (which I couldn't find mention of the size, and this should be included in this caveat section), as well as the potential influence of differential sampling amounts on microbial richness. The caveat section should also clarify that the paper's methods focus on bacteria, due to the 27F/1492R primer initial amplification, but that some archaea were also included due to the second PCR primer set. There indeed caused a bias towards bacteria in this dataset.

Other minor details:

- Fix the abbreviations for nitrate, phosphate and silicate (e.g., NO₃⁻, PO₄³⁻).
- Analyses are DNA-based, so '16S ribosomal RNA gene' should be used or '16S rDNA' terminology used throughout.
- Some coral genera names were not italicized
- Line 63, consider replacing 'large diversity of reef microorganisms' with 'large richness of reef microorganisms compared to other environments'. It seems like this analysis was done using richness, rather than a suite of diversity parameters.
- Line 501/502 – it was confusing that the 18S rRNA gene (add gene on this line) analysis was listed as available. In your rebuttal you mention that this analysis was completed. Why not be clear here that the analysis was completed? If it was not completed, then remove this statement.
- In the rebuttal, it is not ideal to refer the reviewer to another manuscript in bioarchive. Instead, please address the questions/point in the manuscript directly or add the requested info to the supplementary question.

Reviewer #3 (Remarks to the Author):

I was asked to review this manuscript AND the "response to reviewers" but was not involved in the initial review. I do feel that the authors have done an adequate job responding to the reviews. I think that as a summary of a massive sampling effort this manuscript deserves to be published, but I have a number of stern critiques of the revised version that I think will help address both the prior reviewers concerns and address some lingering fundamental problems with the manuscript. Some of my recommendations will have significant impacts on the conclusions drawn from the study, and should be considered carefully, but I feel these analyses address weaknesses in the existing analyses.

Note that I have restricted my comments only to major, significant issues (separated into 5 Points below), so although I have referenced lines in some cases these are not meant as minor edits.

Point 1:

For a paper summarizing the microbiomes of corals in the context of plankton and fishes across the entire Pacific, it is surprisingly hard to look at the figures and get any idea of the composition of the coral microbiomes, the dominant taxa in them, how they differ by species and how they differ from planktonic communities. Basically the fundamental data of the manuscript are hidden in favor of higher-level diversity analyses that are obfuscating and, as raised by all reviewers, not terribly compelling.

Instead, I would note the need for this above other graphics and I would advocate that Supplementary Figures 2, 3, 4, 12 and 14 are the most informative summaries of the patterning and composition of microbial communities, and these should be combined to form Figure 5 (there is no Figure 5. It seems it was removed to supplements as recommended by a reviewer) . Seems like a quick fix to vastly improve the manuscript.

Point 2:

L548-551 The nested PCR on host tissues to enrich prokaryotic DNA needs to be detailed, particularly the cycles devoted to the first and second PCR amplifications, as there is potential for significant bias which would invalidate the conclusions.

Point 3:

L 137-138: The colors don't seem to match the legend. This is a key detail which needs clarification. Are the plankton size fractions in purple in the density plots and/or the NMDS? It looks here like the plankton overlap with the fishes on the NMDS. This is very confusing.

Point 4:

The biggest change I would recommend is a re-analysis of the "community clusters":

Figure 4 is wasted without a clear legend indicating taxa comprising each "community cluster" module. A simple annotation with colors and a list of the dominant taxa within each cluster would be an acceptable addition.

L603-605: As raised by other reviewers, there is no supplementary figure showing the patterns of community clustering to form these "community clusters". The "Review Figure 3" is this type of figure, and adding a biclustered heat map to Review Figure 3 with taxonomic annotations would allow this to be an informative figure: one for each coral coral species could be added to the supplements.

Note that this type of clustering should first be "standard scored" (using something like a z-score) to avoid clusters being driven by relative abundance (cluster 1 is abundant taxa, cluster 2 is rarer taxa, etc.) rather than differential abundance (cluster 1 is taxa enriched on coral sample set a, cluster 2 is taxa enriched on coral sample set b, etc.)

Point 5:

Finally, Figure 6 is not a useful depiction of the data. The results and discussion about the analyses done in Figure 6 and associated supplemental figures are not well-matched to the visuals in Figure 6. I would move Figure 6 to supplement and let that final section of the results and discussion focus on supplemental figures alone, or I would restructure figure 6 to illustrate what is actually discussed (namely, the most abundant and prevalent microbial taxa on each coral and whether each coral has

Reviewer #1 (Remarks to the Author):

The 18S analysis is not detailed here, and I do not have access to the Hume dataset, which could be where this analysis is (line 503, reference 68), or is it reference 34 as stated line 568? Apart from this, I believe the Lombard et al paper details your sampling. Here is where things do not add up in this paper with your response to my comment: section 2.1 of the Lombard et al. paper (lines 205-211) states that the samples collected for the analysis presented in the manuscript I am currently reviewing were SCUBA-3X10, for which photos were taken. Then, section 2.9.1 (lines 328 to 345) states that for these samples, taxonomic identification was done based on morphology (CTAX). It appears that the only 18S analysis was done for the SCUBA-SURVEY samples, which were used for coral host diversity, not for the present manuscript. Nowhere is it said that the SCUBA-3X10 were a subset of the SCUBA-SURVEY. Therefore, it looks like the samples collected for the present study were only assigned taxonomy through pictures and morphological observation, not 18S. Maybe it's all in the zenodo files but please clarify.

-We apologise if the information for this complex sampling design was not clear enough. The coral samples used in the present study were obtained from the [SCUBA-3X10] sampling event and labelled [CTAX] for morphological taxonomic identification, and [CS4L] for metabarcoding. The term metabarcoding includes 16S and 18S, but we now see that it's not specified in Lombard et al. We will ask our colleague to add the information to the Lombard et al. proofs before publication. The metabarcoding sequencing protocols applied to samples [CS4L] are, nevertheless, presented in Table 1 of Belser et al. We have added the information to our method section (lines 670-671 of the track changes version). We have also added the term [SCUBA-3X10] to the methods (line 630), and clearly specified that the 18S analysis is presented in Hume et al. (line 628).

We would like to emphasise that SCUBA-3X10 is not a subset of the SCUBA-SURVEY, which includes different coral species that are not all sequenced yet and not presented in the present paper.

2.2 Best case scenario, even if you were able to identify the genus of each coral with 18S, your choice of using the term “genus” rather than “species” does not address the problem; because you are actually not analysing your data at the genus level (which makes your statement “our study was conducted at the genus level”, line 465, incorrect). Using the term “genus” creates the opposite flaw as using “species”, because you are now assuming that the morphologies that you collected, which were very openly and intentionally restricted to three very specific ones (with the hope of catching only three species altogether), host enough species that they are representative of 3 entire genera. The only way that using “genus” would be correct is if you had done your analysis at the genus level indeed, that is if you had sampled at the genus level, that is if you had sampled multiple species; enough species, in fact, so that collectively they would be representative of each of the genera. If you had done this, you could have said “Within genus X, we found that much diversity”. But given how you sampled, as much as you cannot say that you only have one species per genus, you cannot extrapolate the microbial diversity that you see in corals

showcasing the morphologies of *Millepora platyphylla*, *Porites lobata* and *Pocillopora meandrina* to the entire genera *Millepora*, *Porites* and *Pocillopora*, respectively. Granted, *Millepora* only has a couple of species, but this is not at all the case for the other two genera. That is why you are not doing your analysis at the genus level. You might be doing it at the species level, but you don't know that. You are actually doing this analysis at the "morphology level".

What is done is done, all you can do now is address the problem. To do so, I recommend this: you need to use the term "morphology" instead of "species" or "genus". This is the only accurate way of naming what you're looking at. I advise using something like "morph_MPla", "morph_PLob" and "morph_PMea", for example, to nominate your sample types. It is the only way of ensuring accuracy (and minimising future mis-citation).

-We understand the reviewer's concern and followed his/her recommendations. We have replaced genus/genera with morphotype(s) through the entire document. We also write that: "It should be noted, however, that our study was conducted at the morphotype level, and we cannot exclude that some variations could be due to the presence of different species within a same morphotype." (lines 591-596). Finally, we specify in the method section: "Since different coral species within the same morphotype may be difficult to discriminate by eye, we chose to present our results at the morphotype level: morphotype M. platyphylla, morphotype P. lobata, and morphotype P. meandrina. For the ease of comprehension, the morphotypes are referred respectively as Millepora, Porites, and Pocillopora in the paper." (lines 640-644).

New comments about grammar:

Line 78: I would say "of marine ecosystems" (plural rather than singular)

-Corrected.

Line 82: change "diversity of living species" to "species diversity"

-Changed.

Line 85: it's a small thing, but if I read "microorganisms" after reading a paragraph on corals, I think "Symbiodinaceae". It's a bit strange to me that there is no mention of it in the introduction. But I understand that you did not analyse this here so it's fine if you'd rather ignore them altogether. Just a personal comment I guess.

-Ok.

Line 92: replace "as it was often assessed" by "as it has often been assessed"

-Replaced.

Line 94: replace "recent attempts of counting" by "recent attempts at counting".

-Replaced.

Line 108: remove "also" (because you already have "moreover").

-Done.

Line 186: "The most abundant ASV belonged to the family Endozoicomonadaceae...". Since you're talking about abundance here, please tell us how abundant it was (even if you just give max and min across all morphologies).

-Added (line 216).

Line 382: change "our accumulation curve suggests" with "our accumulation curve shows", because there is no ambiguity in this result.

-Changed.

Reviewer #2 (Remarks to the Author):

This is an unprecedented dataset of coral reef organismal and seawater biomes that provides novel and surprising insights into the bacterial diversity of these biomes. I am supportive of the presentation of the data and findings, and find the paper well written.

I'm surprised that plankton microbial diversity was higher than the animal-based diversity. This contrasts with my view of these systems. Although, now I realize that many studies do not directly compare richness/diversity comparisons between seawater and coral and instead focus on beta diversity and compositional parameters. I don't have an ask for the reviewers here, just a comment.

-We appreciate the comment of the reviewer and are thankful for his/her positive response.

There was just one thing that I think the authors could improve upon, which would strengthen the acceptance of this paper's findings by the larger microbiology community. The paper presents non-identical methods for sample storage (deep frozen vs. preservation fluid) and processing (different DNA extraction methods), yet ultimately compares all data similarly. I understand that the authors had a trade off in terms of sample preservation and processing strategies. Additionally, some sample types (like the seawater analysis) seemed to be tied to methods from prior global analyses, and thus different preservation and DNA extraction methodologies were used. To acknowledge this potential bias in the data, the authors should include a short caveat section that addresses the methodological limitations directly.

Additionally, this section should address the potential microbial abundance/biomass differences between the seawater (50 L of water!) and coral tissue pieces (which I couldn't find mention of the size, and this should be included in this caveat section), as well as the potential influence of differential sampling amounts on microbial richness. The caveat section should also clarify that the paper's methods focus on bacteria, due to the 27F/1492R primer initial amplification, but that some archaea were also included due to the second PCR primer set. There indeed caused a bias towards bacteria in this dataset.

-Following the reviewer's comment we have now added a section on the possible limitations of the method. We now write: "Finally, the comparison of the diversity may have been influenced by the different sample preservation and extraction methods used for the different biomes (see the material and methods section). The number of bacterial cells also probably differed between samples due to differences in sample material (volume of water vs coral material vs fish skin and gut). However, the fact that the rarefaction curve was close to saturation for all biomes, and for all sample types within biomes, indicates a good coverage of the microbial diversity. It suggests that possible methodological biases were limited when doing inter-biome comparisons. It should also be noted that these comparisons targeted mainly bacteria (and not archaea) because of the use of bacterial specific primers in the first PCR step." (lines 589-597 in the track changes version)

-A total of ca. 4g of coral was used for DNA extraction. We have added the information to the method section (line 671). It's also available in Lombard et al.

Other minor details:

- Fix the abbreviations for nitrate, phosphate and silicate (e.g., NO₃⁻, PO₄³⁻).
- Corrected.
- Analyses are DNA-based, so '16S ribosomal RNA gene' should be used or '16S rDNA' terminology used throughout.

- *Corrected to 16S rDNA.*
- Some coral genera names were not italicized.
- *Thank you for noticing. It's now corrected.*
- Line 63, consider replacing 'large diversity of reef microorganisms' with 'large richness of reef microorganisms compared to other environments'. It seems like this analysis was done using richness, rather than a suite of diversity parameters.
- *We followed the reviewer's recommendation and changed the sentence accordingly.*
- Line 501/502 – it was confusing that the 18S rRNA gene (add gene on this line) analysis was listed as available. In your rebuttal you mention that this analysis was completed. Why not be clear here that the analysis was completed? If it was not completed, then remove this statement.
- *We now write that it was completed and specify that outlying samples were removed.*
- In the rebuttal, it is not ideal to refer the reviewer to another manuscript in bioarchive. Instead, please address the questions/point in the manuscript directly or add the requested info to the supplementary question.
- *Noted. We will do so.*

Reviewer #3 (Remarks to the Author):

I was asked to review this manuscript AND the "response to reviewers" but was not involved in the initial review. I do feel that the authors have done an adequate job responding to the reviews. I think that as a summary of a massive sampling effort this manuscript deserves to be published, but I have a number of stern critiques of the revised version that I think will help address both the prior reviewers concerns and address some lingering fundamental problems with the manuscript. Some of my recommendations will have significant impacts on the conclusions drawn from the study, and should be considered carefully, but I feel these analyses address weaknesses in the existing analyses.

-We have carefully considered all the reviewer's remarks and accepted all of his/her suggestions. The constructive and precise comments helped us improve the quality of the manuscript.

Note that I have restricted my comments only to major, significant issues (separated into 5 Points below), so although I have referenced lines in some cases these are not meant as minor edits.

Point 1:

For a paper summarizing the microbiomes of corals in the context of plankton and fishes across the entire Pacific, it is surprisingly hard to look at the figures and get any idea of the composition of the coral microbiomes, the dominant taxa in them, how they differ by species and how they differ from planktonic communities. Basically the fundamental data of the manuscript are hidden in favor of higher-level diversity analyses that are obfuscating and, as raised by all reviewers, not terribly compelling.

Instead, I would note the need for this above other graphics and I would advocate that Supplementary Figures 2, 3, 4, 12 and 14 are the most informative summaries of the patterning and composition of microbial communities, and these should be combined to form Figure 5 (there is no Figure 5. It seems it was removed to supplements as recommended by a reviewer) . Seems like a quick fix to vastly improve the manuscript.

-Following the reviewer's recommendation, we have now grouped the above mentioned supplementary figures into one main figure. Since the results are called early in the paper, this new figure is now named Fig. 2.

Figure 2. Bray-Curtis based MDS ordinations showing differences in microbial community composition within each biomes. a Between Millepora, Porites and Pocillopora and b their overall community composition for the 10 most abundant bacterial orders. c Between

Zanclus cornutus and *Acanthurus triostegus* gut and mucus. d Between *Pocillopora* microbial communities and free-living planktonic communities (size 0.2 – 3µm) sampled close to the *Pocillopora* colonies (colony water). e Between planktonic communities sampled from sea surface water near the islands, surface water over the colonies, and close to the colonies (colony water) for the 0.2 – 3 µm size fraction and f for the 3 – 20 µm size fraction.

Point 2:

L548-551 The nested PCR on host tissues to enrich prokaryotic DNA needs to be detailed, particularly the cycles devoted to the first and second PCR amplifications, as there is potential for significant bias which would invalidate the conclusions.

-To minimise biases, the PCR protocol included a smaller number of cycles for the first PCR step (20 cycles) and a higher number for the second step (25 cycles). We have added the information to the method section (lines 688-690 of the track changes version). We also added that the full PCR protocol is available in the Supplementary Table 4 of Belser et al. (lines 690-691).

Point 3:

L 137-138: The colors don't seem to match the legend. This is a key detail which needs clarification. Are the plankton size fractions in purple in the density plots and/or the NMDS? It looks here like the plankton overlap with the fishes on the NMDS. This is very confusing.

-The colours for plankton and Zanclus were switched on the MDS. Thank you for noticing it. The error occurred when we re-draw the figure following one of the first reviewer's comment. It's now corrected.

Point 4:

The biggest change I would recommend is a re-analysis of the "community clusters":

Figure 4 is wasted without a clear legend indicating taxa comprising each "community cluster" module. A simple annotation with colors and a list of the dominant taxa within each cluster would be an acceptable addition.

L603-605: As raised by other reviewers, there is no supplementary figure showing the patterns of community clustering to form these "community clusters". The "Review Figure 3" is this type of figure, and adding a biclustered heat map to Review Figure 3 with taxonomic annotations would allow this to be an informative figure: one for each coral coral species could be added to the supplements.

Note that this type of clustering should first be "standard scored" (using something like a z-score) to avoid clusters being driven by relative abundance (cluster 1 is abundant taxa, cluster 2 is rarer taxa, etc.) rather than differential abundance (cluster 1 is taxa enriched on coral sample set a, cluster 2 is taxa enriched on coral sample set b, etc.)

-We appreciate the reviewer's comment and changed our figures accordingly. We have added a panel (d) to Fig. 4 (Fig. 5 in the new version) that shows the community composition at the order level for the different community clusters of the 3 corals. We have also added supplementary heatmaps for each coral showing the pattern of community clustering (Supplementary Figure 5). The heatmaps represent the scaled abundance (z-score) of the 20 most abundant orders in each community cluster for each coral.

Figure 5d. Taxonomic composition at the order level of the community clusters. Cluster name colours correspond to the colours of the community clusters in the pie charts.

Supplementary Fig. 5. Heatmap showing the clustering of coral samples (rows) against the 20 most abundant bacterial orders (columns) for each coral morphotype: **a** Millepora, **b** Porites, and **c** Pocillopora. The community cluster colour code corresponds to the ones in Fig. 5.

Point 5:

Finally, Figure 6 is not a useful depiction of the data. The results and discussion about the analyses done in Figure 6 and associated supplemental figures are not well-matched to the visuals in Figure 6. I would move Figure 6 to supplement and let that final section of the results and discussion focus on supplemental figures alone, or I would restructure figure 6 to illustrate what is actually discussed (namely, the most abundant and prevalent microbial taxa on each coral and whether each coral has

-We have followed the recommendation and moved the figure to the supplementary section. The section now only refers to the Supplementary Table 2, where the detailed occurrence of ASVs is given, and to the supplementary Fig. 11 (old Fig. 6).

Please note that the last sentence of the reviewer's paragraph was cut.

REVIEWERS' COMMENTS

Reviewer #1 (Remarks to the Author):

It is looking quite nice now.

Last comments:

1. Belser et al. has not been peer-reviewed. To ensure that what you are publishing here is as solid as possible, please rephrase line 630 of the track-changed version as follows: "Sampling protocols are described in Lombard et al. (2022)67, which includes details of the coral samples used in the present study (SCUBA-3X10, protocols CTAX and CS4L in particular)".

2. Line 119 of the track-changed version: change "three coral species that belong to distinct clades" for "three coral morphotypes that belong to distinct clades".

3. Figure 6: change "genera" for "morphotypes".

Congratulations again for pulling this off!

Reviewer #3 (Remarks to the Author):

The authors have nicely attended to my suggested edits and I think that the manuscript is now suitable for a broad audience appropriate to Nat Comm

Reviewer #1 (Remarks to the Author):

It is looking quite nice now.

Last comments:

1. Belser et al. has not been peer-reviewed. To ensure that what you are publishing here is as solid as possible, please rephrase line 630 of the track-changed version as follows: “Sampling protocols are described in Lombard et al. (2022)⁶⁷, which includes details of the coral samples used in the present study (SCUBA-3X10, protocols CTAX and CS4L in particular)”.

-Done (line 452).

2. Line 119 of the track-changed version: change “three coral species that belong to distinct clades” for “three coral morphotypes that belong to distinct clades”.

-Done.

3. Figure 6: change “genera” for “morphotypes”.

-Done.

Congratulations again for pulling this off!

Reviewer #3 (Remarks to the Author):

The authors have nicely attended to my suggested edits and I think that the manuscript is now suitable for a broad audience appropriate to Nat Comm